



# Profiling float observation of thermohaline staircases in the western Mediterranean Sea and impact on nutrient fluxes

Vincent Taillandier[1], Louis Prieur[1], Fabrizio D'Ortenzio[1], Maurizio Ribera d'Alcalà[2,3], Elvira Pulido-Villena[4]

[1] CNRS, Sorbonne Universités, Laboratoire d'Océanographie de Villefranche, UMR7093, Villefranche-sur-Mer, France
[2] Department of Integrative Marine Ecology, Stazione Zoologica Anton Dohrn, Napoli, Italy
[3] Istituto per lo Studio degli Impatti Antropici e Sostenibilità in Ambiente Marino, CNR, Roma, Italy
[4] Aix-Marseille Université, CNRS, Université de Toulon, IRD, Mediterranean Institute of Oceanography, UMR7294, Marseille, France

*Correspondence to*: Vincent Taillandier (taillandier@obs-vlfr.fr)

**Abstract.** Characterizing the spatio-temporal arrangements of inorganic nutrients is critical to improve our understanding of the marine biological primary production. Among the processes contributing to nutrient distributions, diapycnal diffusion plays a crucial role for the supply of nutrients to the surface productive zone, and for the equilibration of vertical differences in nutrient concentrations induced by large scale thermohaline circulation. This is the case in the western Mediterranean Sea, where Levantine intermediate waters (LIW), that circulate below the surface layer, regionally distribute the nutrient stocks conveyed from the eastern basin or provided by terrestrial inputs, atmospheric deposition, and remineralization of organic matter. In the present study, we focus on the role played by diffusive processes in the LIW fertilization, considering long-term observations of thermohaline staircases. In association with the unprecedented contribution of profiling floats to explore their structural changes, the fine characterization of western Mediterranean thermohaline staircases sampled during the cruise PEACETIME can be carried out from a different perspective. Observations revealed that thermohaline staircases develop over epicentral regions confined inside large scale circulation features and sustained by saltier LIW inflows on the periphery. As observed in the Algerian Basin, these epicentral regions are thought to be site of active mixing, with changes of seawater properties by about +0.06°C in temperature and +0.02 in salinity during the four years of observation. In-situ lateral density ratios are analysed in the view of theoretical predictions to identify and untangle i) salt fingering as driver of water mass conversion, with ii) isopycnal diffusion as spreader of heat and salt from the surrounding sources. In the Tyrrhenian Sea, the resulting nutrient fluxes bring upward from deep waters 5 µmol/m²/d in nitrate, which represents one fourth of LIW fertilization by diapycnal diffusion, but remains a secondary contributor to the enrichment of Ionian water inflows.

## 1. Introduction

In most regions of the ocean, the accumulation of phytoplanktonic biomass is limited by the availability of nutrients (Eppley and Peterson, 1979). This is the case of the Mediterranean Sea, an ultra-oligotrophic basin where primary production is



generally weak because nutrient fluxes into the sunlit surface layers are very low during most of the year. Apart from dense water formation zones where spring phytoplankton blooms are observed (D'Ortenzio and Ribera d'Alcalà, 2009), nutrient fluxes favoring phytoplankton uptake occur at the level of the thermocline (Pasqueron de Fommervault et al., 2015) and

ubiquitous deep chlorophyll maximum develops in sub-surface (Lavigne et al., 2013; Barbieux et al., 2019).

Similarly to the global ocean (Williams and Follows, 2003), the Mediterranean nutrient stocks below the thermocline are arranged by hydrodynamical transport; they are distributed at large scales by thermohaline circulation cells (Wüst, 1961). In the western Mediterranean, deep waters (hereinafter DW) result from winter convection in the Provençal Basin and in the Ligurian Sea (Medoc Group, 1970; Prieur et al., 1983). Likewise, the eastern Mediterranean DW are formed in the southern

Adriatic Sea and intermittently in the Aegean Sea (Lascaratos et al., 1999; Roether et al., 2007). Mediterranean intermediate waters (hereinafter LIW) are, instead, mostly of Levantine origin and result from shallow convection in different sites of the Levantine Basin further subducted under the surface layer (Nittis and Lascaratos, 1999; Malanotte-Rizzoli et al., 2003). Overall, two distinct thermohaline circulation cells act in. Those involving DW are exclusive of the eastern or the western Mediterranean basins, whereas the one driven by LIW encompasses the whole Mediterranean. They regionally distribute the

nutrient stocks, whether by DW that are spread from Liguro-Provençal Basin and from Adriatic-Aegean Seas, or by LIW that are conveyed from the eastern Mediterranean (Ribera d'Alcalà et al., 2003; Kress et al., 2003).

Interestingly, the preformed nutrient concentrations in LIW are very low or close to zero (Pasqueron de Fommervault et al., 2015) while they get enriched along the path, becoming a relevant storage of nutrients in the western Mediterranean. The nitrate concentration is about 5 μmol/kg at the Strait of Sicily, which is already quite high compared to the intermediate

concentrations of the eastern Mediterranean, but it is one half of the concentration measured in the western Algerian Basin (Pujo-Pay et al., 2011). In order to reach such concentrations in the Ionian water inflows, LIW would export the major part of atmospheric and terrestrial inputs from the eastern Mediterranean (Ribera d'Alcalà et al., 2003). On the other hand, the mechanisms of LIW fertilization among the western Mediterranean are not so clear. External sources (river and coastal runoffs or atmospheric deposition) are even larger than the eastern basin, but the remineralization of organic matter settling from the

surface layer becomes a major contributor there (Béthoux et al., 1998). It is worthwhile to note that the LIW enrichment in the Tyrrhenian Sea is comparable to the Provençal Basin, as documented by the cruise PEACETIME (Figure 1). This observation has raised our interest on the contribution of alternative processes that could explain such regional modifications in nutrient stocks.

Although the general arrangement of nutrients is driven by large scale circulation features, internal processes acting on the

vertical scale could modify the distribution of inorganic matter. This is the case of diapycnal diffusion, which can be particularly efficient at long terms when it is enhanced by the vertical mixing process of salt fingering (Hamilton et al., 1989; Fernandez-Castro et al., 2015). Under appropriate conditions reviewed by Schmitt (1994), salt fingering tends to organize the water column into series of mixed layers separated by sharp temperature and salinity gradients, ultimately designing thermohaline staircases. Since this natural phenomenon has been elucidated (Stern, 1960; Stern and Turner, 1969), the western

Mediterranean Sea has become one of the world's locations for in situ characterization of the resulting step-layer structures.



Molcard and Tait (1977) reported the presence of persistent staircases in the central Tyrrhenian Sea, composed of 10 homogeneous layers between 600 m and 1500 m, with constant sea water properties (at the instrumental precision of 0.01°C in temperature and 0.01 in salinity) over three years of observation. Later, Zodiatis and Gasparini (1996) described the areal extent of the structure, covering large distances from the central area, progressively weakening and disappearing near the

coasts. Krahmann (1997) reported the first synoptic observations of thermohaline staircases in the Algerian Basin, confirmed by Bryden et al. (2014) from biannual surveys across this region. More recently, Buffett et al. (2017) revealed their remarkable spatial continuity from very high-resolution synoptic observations.

Staircases are thought to be sites of thermohaline changes among well-organized homogeneous layers, when such singular structures are maintained over long terms. Changes can either take the form of intrusions of heat and salt spreading horizontally

inside layers (McDougall, 1985; Merryfield, 2000), or slow temporal trends induced by the downward flux of heat and salt between layers (Schmitt, 1994; Radko and Smith, 2012). In order to untangle the two processes, an accurate measurement of thermohaline changes requires that the temporal continuity of layers is effective, which is the most challenging aspect to access from observations. Indeed, because thermohaline staircases occur far from the coastal zones and they develop in the deep interior oceans, sampling capability is often limited whether to high resolution but short-term records (e.g. Buffett et al., 2017),

or coarse temporal resolution but long-term records (e.g. Falco et al., 2016; Durante et al., 2019). Moreover, most of the existing studies have not considered the role of thermohaline staircases in the nutrient distribution, because pertinent observations of biogeochemical parameters were hard to be obtained in the water column at relevant temporal and spatial scales. To this concern, profiling floats provide valuable datasets (that can include biogeochemical parameters) to suitably explore the characteristic scales of these features. These autonomous platforms drift in the interior ocean and evenly surface

for positioning and data transmission, with typical sampling rates of some days during periods of some years.

In the present study, we propose to focus on the role played by diapycnal diffusion in the equilibration of nutrient concentrations in the western Mediterranean Sea considering nutrient fluxes between DW, LIW and surface waters, given the regional characteristics of the mixing process. We use the dataset acquired during the cruise PEACETIME (Guieu et al., this special issue), that carried out a large scale survey of the western Mediterranean Sea in May-June 2017, although data of earlier

cruises are also considered. Shipboard data are combined with observations obtained by a unique array of Biogeochemical Argo floats deployed in the western Mediterranean Sea (D'Ortenzio et al., 2019). The progression of this study will be i) to provide a fine characterization of the staircases recorded by shipboard data, ii) to infer their spatial extension and temporal persistence using profiling float observations, iii) to confirm the salt fingering activity and iv) to assess the contribution of this mixing process in the progressive enrichment of LIW nutrients.



## 2. Data and methods

### 2.1 CTD profiles

During the cruise PEACETIME in May-June 2017 (Guieu and Desboeufs, 2017; Guieu et al., this special issue), a Conductivity-Temperature-Depth (CTD) underwater unit was lowered from the surface to the bottom. Pressure, in situ temperature and conductivity of sea water were measured using SBE911+ CTD. This instrumental package provides continuous acquisitions at 24 scans per second. The depth of each scan is scaled in pressure (unit: bar). Raw data are processed into quality-controlled profiles of salinity scaled in practical salinity units and potential temperature referenced to surface (0 dbar), at the vertical resolution of 1 dbar (about 1 m). For sake of simplification, salinity is now referred to the parameter "practical salinity" (no dimension); temperature is now referred to the derived parameter "potential temperature" (unit: °C). The accuracy of measurement of the CTD unit is about 1 dbar in pressure, 0.001°C in temperature and 0.003 in salinity. The PEACETIME shipboard dataset is complemented with CTD profiles collected during three earlier cruises with same instrumental package, MedSeA (Ziveri and Grelaud, 2015) in May 2013, SOMBA-GE (Mortier et al., 2014; Keraghel et al., 2019) in August 2014, and BioArgoMed (Taillandier et al., 2018) in May 2015.

Another CTD dataset has been collected by profiling floats, autonomous platforms that drift in the interior ocean and evenly surface for positioning and data transmission. The profiling floats considered in this study belonged to the NAOS Biogeochemical (BGC) Argo array (D'Ortenzio et al., 2019). The sampling strategy of this array is well suited for observations of thermohaline staircases, rather than the MedArgo array (Poulain et al., 2007). Indeed, the BGC-Argo profiles are deeper, 1000 dbar instead of 700 dbar. Second, the data collections have limited spatial dispersion and remain longer inside basin of deployment, thanks to a parking depth of 1000 dbar, instead of 350 dbar for MedArgo array. Third, BGC-Argo floats have higher vertical resolution between 250 and 1000 dbar, 10 dbar instead of 25 dbar for MedArgo array. These floats are equipped with SBE41CP pumped CTDs, sensors of high stability adapted from mooring applications, that provide continuous acquisition at 0.5 Hz with instrumental precision of 0.01 for salinity, 0.002°C for temperature, 2.4 dbar for pressure (Wong et al., 2019). CTD profiles are collected during ascent from parking depth to surface, lasting about three hours with a nominal vertical speed of 0.1 m/s. In the layer 250-1000 dbar, each record is an average of temperature and salinity inside 10 dbar slices (about 200 scans), which reduces the noise of raw acquisitions.

The selected BGC-Argo profiles were collected inside two regions: (8°E-16°E, 38°N-42°N) in the Tyrrhenian Sea, (2°E-9°E, 36°N-40°N) in the Algerian Basin (Figure 1). The resulting time series of CTD profiles lasted four years between May 2013 and May 2017 (date of the cruise PEACETIME) in both areas (Table 1), with a nominal time resolution of seven days that can increase up to one day. In the Tyrrhenian Sea, a set of 323 CTD profiles is collected by two float deployments. The first float 6901491 was deployed in May 2013 during the cruise MedSeA, it has been recovered two years after in the south west sector of the basin during the cruise BioArgoMed while it was still active. The second float 6901769 was deployed in the continuation of the float 6901491, at the same date and location than the recovery; it left by the Sardinian Channel two years after and it was lost in January 2018. In the Algerian basin, a set of 336 profiles is collected by three float deployments. The first float



6901513 was deployed in May 2013 during the cruise MedSeA, it left the Algerian basin more than two years after and it was recovered in June 2016 in the Provençal Basin during the cruise MOOSE-GE (Testor et al., 2010). The second float 6902732

has been refitted from its previous deployment in the Tyrrhenian Sea and it has been deployed in the continuation of the float 6901513, at the same date and location than the recovery; it entered the Algerian basin offshore Minorca in January 2017 and left the basin one year after. The third float 6901600 was deployed in August 2014 during the cruise SOMBA-GE, it was lost in the western part of the basin after more than one year of operation.

Overall, we consider in this study a dataset of about 700 profiles, acquired by BGC-Argo floats at 10-dbar vertical resolution

and shipboard CTD package at 1-dbar vertical resolution, with a metrological harmonization at the precision of the BGC-Argo standards (0.002°C in temperature and 0.01 in salinity). The systematic metrological verification of BGC-Argo CTD sensors, comparing their first profile at deployment with concomitant shipboard CTD profile, confirmed the absence of initial calibration shift for the five floats. This metrological verification has been done also for the recovered floats (6901513, 6901491), which confirmed the absence of temporal drift larger than the nominal uncertainties of CTD measurements.

**2.2 Thermohaline staircases**

As a preliminary note, the terminology used in the text is illustrated in Figure 2 for sake of clarity. Thermohaline staircases are observed where large scale temperature and salinity fields decrease with depth in a manner that favors the mixing process of salt fingering (Schmitt, 1994). This natural process has been predicted by theoretical models and reproduced by laboratory experiments and numerical simulations (Schmitt, 1994; Stern and Turner, 1969; Merryfield, 2000; Radko et al., 2014). Salt

fingers take the form of tinny cells (some centimeters wide, some tens centimeters tall) across which the rising and sinking fluids mostly exchange heat, as thermal molecular diffusivity is larger than that of salt by two orders of magnitude. In the vertical extension of the cells, sinking (resp. rising) fluids find themselves saltier (resp. fresher), but with same temperature, than waters at the same depth. The resulting buoyancy instability ends up to drive convection in the adjoining mixed layers. When salt fingering is active, the whole transition zone between the warm and salty waters and the cold and fresh waters will

be reorganized into mixed layers separated by temperature and salinity steps, until an equilibrium of well-developed staircases is reached (Radko, 2005). The relative stability of this arrangement, whether across a single step (local) or considering the whole transition zone (bulk), can be expressed in terms of density ratio

$$R_\rho = (\alpha . \partial\theta/\partial z) / (\beta . \partial S/\partial z) \tag{1}$$

which relates the stabilizing vertical temperature ($\theta$) gradient and the destabilizing vertical salinity (S) gradient. $\alpha$ and $\beta$ are

the thermal expansion and haline contraction coefficients of seawater referenced to the same pressure than potential temperature

$$\alpha = -(1/\rho) . \partial\rho/\partial\theta \quad , \quad \beta = (1/\rho) . \partial\rho/\partial S \tag{2}$$

where $\rho$ is the potential density derived from pressure, temperature and salinity using the equation of state of seawater. Low values of density ratio are conditional for staircase formation, between 1 and 1.7 with respect to field observations (Schmitt et



al., 1987). As an essential prerequisite for staircase formation in the Tyrrhenian Sea and the Algerian Basin, the probability

distribution of local density ratios fits to low conditional values (1-1.7) under the LIW core (Onken and Brembilla, 2003).

The detection of thermohaline staircases in the CTD dataset (Section 2.1) is performed by a suite of profile-by-profile

diagnostics built by the theoretical elements presented above in this Section 2.2. In the first stage of the method, the depth

range of the transition zone is extracted from each profile, by screening downwards for the depth of the salinity maximum

(LIW) then the depth of the salinity minimum (DW) underneath 250 dbar. The bulk vertical gradients of temperature and

salinity are derived from water properties at the top and at the bottom of the transition zone. The resulting bulk density ratio

($R_\rho$, Equation 1) is checked to be in the range of 1-1.7. If so, the second stage of the method is run: the distribution of pairs

(salinity, temperature) that belong to the transition zone is evaluated by a rapid hierarchical classification algorithm in order

to detect concentration points representative of mixed layers (Jambu, 1981). A concentration point is identified as a set of

successive scans which temperature does not vary by 0.005°C, salinity does not vary by 0.005. In addition, this set of scans

must be composed of at least 3 scans for BGC-Argo profiles, 15 scans for shipboard profiles, limiting the detection to layers

thicker than 30 dbar and 15 dbar respectively. In a third stage of validation, the scans belonging to the concentration points are

superimposed onto the full profile and the vertical alternation of steps and layers is checked by a visual inspection. If so, the

profile is reported as an observation of staircases.

Several diagnostics are run on the set of profiles with staircase detection. The fine structure characterization includes layer

properties (i.e. seawater temperature and salinity at every validated concentration point), interlayer temperature-salinity steps,

associated interlayer density ratios, layer thickness (i.e. number of concentration points times vertical resolution of profiles),

and step thickness (i.e. the depth interval between two adjacent layers). The occurrence of staircase is estimated among the

BGC-Argo dataset by the percent of profiles with at least one concentration point per profile (reported in Table 1). The

continuity of layers is displayed among the shipboard and BGC-Argo datasets by the persistence of some layer properties.

Layers are conventionally numbered using the fine structure characteristics of the profiles collected during the cruise

PEACETIME. Changes of temperature ($\Delta\theta_i$) and salinity ($\Delta S_i$) within every indexed layer (i) are examined among the BGC-

Argo dataset in terms of lateral density ratio

$$R_L^i = (\alpha.\Delta\theta_i) / (\beta.\Delta S_i) \tag{3}$$

$R_L^i$ are determined by least square fits of layer distributions in a temperature-salinity diagram normalized by $\beta/\alpha$.

**2.3 Vertical fluxes of nutrients**

The vertical transfer of nutrients in presence of thermohaline staircases is evaluated through the nutricline (between surface

waters and LIW), and across the transition zone (between LIW and DW). The two components are parameterized as a diapycnal

diffusive flux, written as the product of the vertical diffusivity of salts (K) and the vertical gradient in nutrient concentration

(C)

$$F_c = K.\partial C/\partial z \tag{4}$$



Vertical fluxes are quantified using the discrete profiles of nutrient concentrations collected at every station of the cruise PEACETIME. Dissolved inorganic nitrate and phosphate were determined in seawater samples collected by Niskin bottles at discrete depth levels in concomitance with the shipboard CTD profiles. Concentrations at sub-micromolar level were measured

on board by the standard automated colorimetric method (Aminot and Kerouel, 2007), using a Seal Analytical continuous flow AutoAnalyzer III (AA3).

Across the transition zone, the vertical diffusivity of salts (including dissolved inorganic nutrients) would be controlled by salt fingering. The coefficient K is computed following the formulation of Radko and Smith (2012)

$$K_{sf} = k_T.R_\rho.( a_s / (R_\rho - 1)^{1/2} + b_s )  \text{ with }  a_s = 135.7  \text{ and }  b_s = -62.75 \tag{5}$$

where $k_T = 1.4 \ 10^{-7}$ m$^2$/s is the molecular diffusivity of heat and $R_\rho$ is the bulk density ratio given by Equation 1. Through the nutricline instead, the vertical diffusivity would be evaluated by the turbulent kinetic energy dissipation rate ($\varepsilon$). Considering the Osborn's (1980) relationship with a constant mixing efficiency of 0.2, the coefficient K is written as

$$K_{turb} = 0.2 \ \varepsilon / (-g/\rho_o.\partial\rho/\partial z) \tag{6}$$

where g is the gravitational constant and $\rho_o$ the reference density of seawater. Substituting Equation 6 in Equation 4 yields to

a diffusive flux written as the product of $\varepsilon$ and the gradient of nutrient concentration across isopycnals (Omand and Mahadevan, 2015).

$$F_c = -0.2 \ \rho_o/g. \ \varepsilon . \ \partial C/\partial\rho \tag{7}$$

Vertical fluxes through the nutricline are evaluated following Equation 7. However, there were no measurements of turbulent kinetic energy dissipation rates carried out during the cruise PEACETIME. In lack of concomitant data, we use microstructure

observations collected over several cruises from 2012 to 2014 recently reported in the western Mediterranean Sea (Ferron et al., 2017). Rough values of $\varepsilon$ in the nutricline can be extracted from their regional-averaged vertical profiles over the Algerian basin and the Tyrrhenian Sea. The turbulent kinetic energy dissipation rates sharply decrease in the depth range of nutricline, from values of about $8.10^{-9}$ W/kg at 100 m depth to $7.10^{-10}$ W/kg at 300 m depth. These estimates are in agreement with $\varepsilon$ values reported by Cuypers et al. (2012): between $6.10^{-9}$ W/kg and $10^{-8}$ W/kg in the surface layer (20-100 m).

## 3. Results

### 3.1 Observation of staircases in the Tyrrhenian Sea

During the cruise PEACETIME, a station has been performed in the central Tyrrhenian Sea, a well-characterized deep area where intense thermohaline staircases are confined (Molcard and Tait, 1977; Zodiatis and Gasparini, 1996; Falco et al., 2016). Repeated profiles from the surface to the bottom were collected every day, they show well-ordered thermohaline staircases

(Figure 3). At this short observation timescale, there is a strong reproducibility of the vertical structure. LIW properties remain stable at (14.34°C, 38.82), although with a slight uplift of the salinity maximum (470 dbar at cast 1, 400 dbar at cast 4). DW properties remain equally stable under 2500 dbar at (12.98°C, 38.50). The bulk temperature and salinity gradients are similar





for all the casts, respectively 0.00065°C/m and 0.00015/m. The bulk density ratio ($R_\rho$, Equation 1) is equal to 1.32, which is lower than 1.7, the usual threshold for the development of thermohaline staircases (Section 2.2). The transition zone is occupied

by mixed layers of homogeneous properties (variance close to the instrumental precision) and of various thicknesses (from few meters to some hundred meters at the metric resolution of the profiles). Salinity steps can be sharp or gradually smoothed by small transient layers that split and merge during the four days of observation. For example, between cast 3 and cast 4 (Figure 3), the transient layers disappeared from the step at 1250 dbar and appeared in the step at 1700 dbar, meanwhile the layer in between was lifted by about 20 dbar.

Indeed, as reported in Table 2, the layer and step thicknesses can fluctuate up to 17 dbar during the four days of observation, which are signs of active convection. Instead the temperature-salinity characteristics remain stable. Temperature-salinity steps increase with depth until 956 dbar (step 4/5), then progressively decrease. The three main layers (5, 6, 7) are located between 983 dbar and 1871 dbar depth, with thickness ranging between 184 to 332 dbar. They are associated to large steps of temperature and salinity, by 0.06-0.19°C and 0.02-0.05.

The spatio-temporal extensions of this observation can be inferred by two BGC-Argo deployments in the Tyrrhenian Sea that preceded the cruise PEACETIME. This dataset provides a continuous observation of the vertical structure in the upper 1000 dbar with a resolution of one to seven days from June 2013 until May 2017 (Section 2.1, Table 1). Temperature and salinity scans displayed in the depth range of 300-1000 dbar reveal three stripes at roughly constant properties (Figure 4). The stripes are heavy concentrations of scans that correspond to well-mixed layers, while light concentrations of scans in between the

stripes correspond to steps. The temperature and salinity values along these stripes are diagnosed by the detection method (detailed in Section 2.2) and analyzed together with the layer properties of the central station (Table 2). The quantification of layer and step thicknesses can be hazardous on BGC-Argo profiles because of limited vertical resolution, smoothing effects of averaged measurements by 10 dbar slices, and reduction of sensing aperture due to limited profile depth that may truncate the lowest detected layer. Thus, only the temporal evolution of the layer properties (temperature, salinity) can be considered from

the BGC-Argo dataset.

As sketched out in Figure 4, thermohaline staircases were observed almost continuously during the four-years period of BGC-Argo collection. The proportion of staircase detections within this collection reaches 79% (Section 2.2, Table 1). This observation extends over the southwest sector of the Tyrrhenian Sea until the Sardinian Channel (Figure 5, left panel), suggesting a unique structure spreading from the central area up to the southwest border. The obtained vertical structure is in

agreement with the ones of shipboard profiles collected during PEACETIME and during float deployments and recovery two and four years before (Figure 5, right panels). Two layers appear quasi-persistent with steady properties (Figure 5, lower right panels). The first one at (13.65°C, 38.67) corresponds to the layer 3 in the numbering of the station PEACTIME (Table 2), the second one at (13.55°C, 38.65) corresponds to the layer 4. The temperature and salinity gradients at step 3/4 remain of same amplitude. In complement to this reconstruction of the structure, the layer 2 was detected during two large periods of about

one year, the layer 1 during a period of six months, the layer 5 during a short period of three months. As a result, changes on top of the transition zone can whether reduce the number of layers, when LIW are relatively less salty and less warm (second



half 2015), or increase the number of layers, while LIW get warmer and saltier (second half 2016). Changes of layer average depth similarly modulate the number of layers, due to vertical translations of the whole structure with the LIW core (Figure 5, upper right panel).

### 3.2 Observation of staircases in the Algerian Basin


Regarding the Algerian Basin (Figure 6), the CTD dataset covers most of the abyssal plain where thermohaline staircases are able to develop (Krahmann, 1997; Bryden et al., 2014). The cruise PEACETIME provided a zonal transect between 1°E and 8°E with coarse spatial resolution. Three BGC-Argo floats complemented this transect covering the central basin with an average speed of 4 km per day (up to 10 km per day), which is twice larger than that of float motions in the Tyrrhenian Sea.

Such a large dispersion in the deep layers (1000 dbar) can be ascribed to a vigorous basin-scale barotropic circulation, characterized by two permanent cyclonic gyres, and delineated by closed f/H contours (Testor et al., 2005). The area of staircase detection (67% of the profiles, Table 1) seems to be shaped by these contours, marking their preferential development inside the so-called Algerian Gyres. More precisely, there is a systematic detection inside the box (37°20'N – 38°N, 4°E – 6°E), whereas staircases were more sporadically observed everywhere else.

During the cruise PEACETIME, four stations have been performed along the 38°N parallel during which nine casts were collected, including six daily casts repeated at a long station (Figure 7). Thermohaline staircases were observed at every station, in the transition zone from the LIW core (300-500 dbar) down to the DW pool (1400 dbar). They exhibit well-ordered steps and layers in the middle of the transect (long station and cast 8), notably with saltiest observed LIW in cast 8, while the profiles at the edges appear jumbled with no readily apparent pattern. At the long station (casts 2-7), the largest salinity steps can be

sharp or gradually smoothed by small transient layers that split and merge during the first days of observation. The structure is eroded in the last days (casts 6-7 in Figure 7), probably due to the growing influence of a mesoscale eddy sampled at the westernmost station (deeper immersion of the LIW core in cast 1) progressively moving eastwards at that time.

Even with spatial and short-scale temporal fluctuations, the bulk temperature and salinity gradients remained similar for all the casts, respectively 0.00060°C/m and 0.00013/m, as well as the bulk density ratio ($R_\rho$, Equation 1) equal to 1.38. Note that

the bulk density ratio is lower than 1.7, the threshold for the development of thermohaline staircases (see Section 2.2). In a fine description of the vertical structure (Table 3), the transition zone is layered quasi-evenly, with six steps of about 0.05°C in temperature and 0.01 in salinity. The largest step 3/4 separates the two thickest layers (about 70 dbar). Interlayer density ratios decrease from 1.4 to 1.25, with a major variation at the step 3/4.

The spatio-temporal extensions of this observation can be inferred by the three BGC-Argo deployments that preceded the

cruise (Section 2.1). Two separate periods were sampled: from May 2013 until January 2016 then during the whole year 2017 (Table 1). The screening of temperature and salinity scans in the depth range of 300-1000 dbar reveals clear stripes over several periods of some months (Figure 8). In other periods of same duration, the steps are less marked, the concentration of scans between stripes becoming heavier. In comparison with the observation in the Tyrrhenian Sea (Figure 4), the continuity of the layers (stripes) is less clear. The sharpness of the steps is modulated, as depicted in Figure 8 by different levels of scan





concentration between stripes. When the structure is sharp, layer properties can encounter short-term changes; they stabilize while the structure is progressively eroded.

The layer properties have been diagnosed by the staircase detection method detailed in Section 2.2. Their evolution can be depicted considering the shipboard CTD and BGC-Argo datasets (Figure 9). The profiles collected by the floats, although limited to 1000 dbar, can provide an almost full observation of the vertical structure (the first 5 layers over 7 reported in Table 3).

3). They reveal the reproducibility of the layering pattern over years, in a correct continuity with the one documented in their vicinity by ship surveys. Moreover, gradients at the largest step 3/4 are in agreement with the values of the stations PEACETIME (Table 3), about 0.07°C in temperature and 0.02 in salinity. In this temporal view, the temperature and salinity changes within each layer act during distinct events (delimited by red lines in Figure 9), when LIW properties suddenly increase. Then, there is a steady sequence with the stabilization of layer properties, and finally the erosion of steps as the LIW

properties decrease down to a certain threshold (13.35°C, 38.56). Layers properties would get larger in response to the increase of LIW characteristics. This temporal description is further analyzed together with spatial variations, regarding two specific episodes (Figures 8 and 9, red lines).

During a first episode of three months, the float 6901513 drifted westwards profiling every 20 km, along a 250-km zonal transect around the 37°30'N parallel (Figure 10, left panel). In contrast with the middle of the transect, the profiles appear

jumbled at the eastern and western edges (Figure 10, upper right panel). In other words, the temperature and salinity profiles with depth-decreasing values are locally inverted in the depth range of the transition zone, which disrupt the homogeneity of the layers. These local inversions among layers reflect the changes of layer properties reported above (Figure 9), with their slight increase at the beginning of the episode and their decrease at the end. Changes of layer and step thicknesses can be more clearly documented by a representation of profiles aligned at the depth of the step 3/4 (Figure 10, lower panel). This

representation withdraws the depth fluctuations of layers near the step 3/4. Staircases appear well developed at the middle of the transect: steps are sharper and layers are thicker than on the two edges, with the apparition of a small transient layer splitting the step 3/4. This observation suggests well-ordered thermohaline staircases confined inside an epicentral region delineated between 4°15'E and 5°30'E meridians.

During a second episode of four months, the float 6901600 completed a cyclonic gyration, profiling every 10 km along this

path of 60 km radius inside the eastern Algerian Gyre (Figure 11, left panel). The profiles appear jumbled in most of the area crossed by the float (until 22 February 2015, Figure 11 upper right panel), which covers the sector north of the 38°10'N parallel and east of the 5°40'E meridian. In contrast, the layering is sharp and homogeneous for profiles collected in the neighborhood of (37°45'N, 5°20'E). Moreover, layer thicknesses are stable in this epicentral region, they change in the northeast sector (Figure 11, lower right panel). As already described in the first episode, local inversions within layers are associated to changes

of layer properties. The excursion low-high-low of layer temperature-salinity values (Figure 9, right panel between red lines) matches with the increasing then decreasing distance to the location (37°45'N, 5°20'E) (Figure 11, left panel).

The two episodes detailed the spatial extension sketched out in Figure 6, with active well-ordered thermohalines staircases confined inside the eastern Algerian Gyre, and their progressive erosion all around. Moreover, these episodes confirmed the





connection between layers of fluctuating properties, characteristic of spatial variations rather than temporal changes.

Positioning layer properties in a temperature-salinity diagram, values are aggregated by layer along separated lines (Figure 12, upper left panel). The float and the cruise records are distributed from the oldest to the newest along these lines, with a succession in time of float 6901513 (blue), SOMBA-GE (purple triangles), float 6901600 (green), PEACETIME (purple dots), and float 6902732 (red). As a result, these lines document a temporal trend at inter-annual scale, as the five connected layers get warmer by about 0.06°C and saltier by about 0.02 during the four years of observation.

The changes of layer properties, whether at short scales characteristic of spatial variations, or at large scales characteristic of inter-annual trends, are examined in terms of lateral density ratio (Equation 3, Table 4). They are displayed by the slope of layer distributions in the temperature-salinity diagrams (Figure 12). Each layer distribution appears shaped along a line crossing isopycnals, as a composite of segments nearly parallel to isopycnals (Figure 12, upper left panel). The gross lateral density ratio associated to this distribution is in the range of 0.65 – 0.78, with an average of 0.72 (Table 4). Considering the

episodes separately, their distribution is encapsulated in single segments that have slopes close to isopycnals (Figure 12, lower panels). Given the short timescale of each episode (3-4 months), changes of layer properties are attributed to spatial variations. The lateral density ratio is in the range of 0.89 – 0.93 for the first episode, 0.82 – 0.98 for the second episode, with an average of 0.91 in both episodes (Table 4). Considering only the records inside the epicentral region, the distribution extents along lines crossing isopycnals (Figure 12, upper right panel). The segmentation visible with the whole dataset is smoothed with this

limitation, somehow filtering the effects of spatial variations. In this case, the lateral density ratio illustrates the inter-annual trend in the relative changes between layer temperature and layer salinity. This ratio is in the range of 0.74 – 0.83, with an average of 0.80 (Table 4). As detailed further in Section 4.3, these estimations of lateral density ratios conjecture water mass conversion within thermohaline staircases that is driven by two distinct processes, one acting at large spatial scales, the other at large temporal scales.

**3.3 Estimation of nutrient fluxes in presence of thermohaline staircases**

Vertical and lateral fluxes representative of large scale nutrient dynamics over the Mediterranean Sea are examined in the view of the dataset collected during the cruise PEACETIME in May-June 2017 (Section 2.3). Regarding the geographical distribution of nitrate concentrations inside LIW (Figure 1), a progressive increase was observed along their pathway from the Ionian Sea to the Algerian Basin. This enrichment is particularly significant in the Tyrrhenian Sea, with an increase in nitrate

concentration by 2 µmol/kg between the eastern Tyrrhenian record and the southwestern Sardinian record. Instead, nutrient concentrations inside LIW remain homogeneous in the Algerian Basin, after the sharp increase (by 2.5 µmol/kg in nitrate) after crossing the Provençal Basin. They are equally homogeneous in the Ionian Sea, however with lower concentrations (about 3 µmol/kg in nitrate).

The spatial variations of nutrients are more deeply analyzed using five contrasted stations selected along the LIW pathway

(Figure 13): one in the Algerian Basin corresponding to the cast 8 of Section 3.2, one in the Ionian Sea upstream the Strait of Sicily, one in east Tyrrhenian downstream the strait, one in central Tyrrhenian Sea corresponding to the cast 1 of Section 3.1,





and one in the continental slope southwest Sardinia. Young LIW are found in the Ionian station with strong salinity properties (larger than 38.9) at 200 dbar depth, whereas the lowest LIW properties are found in the Algerian station (salinity 38.6 at 450 dbar). The three other stations have transitional properties, down to 38.7 in salinity southwest Sardinia. The nutrient

concentrations follow the inverse progression than salinity in LIW: nitrate profiles at the five stations are clearly differentiated below 250 dbar, showing inflow of low nutrient waters from the Ionian Sea to the eastern Tyrrhenian (same values in nitrate at 450 dbar) and their progressive enrichment until the Algerian Basin.

The contribution of thermohaline staircases in the accumulation of nutrients in the Tyrrhenian Sea can be assessed according to the underlying vertical fluxes. The vertical layout of nutrients is first examined in presence of thermohaline staircases at the

previously described stations (Figures 3 and 7). Note that the cast 1 of the Algerian Basin has been moved aside from the present analysis because its acquisition inside a mesoscale eddy (Section 3.2) provided anomalous nutricline characteristics. The profiles of nutrient concentrations are considered across isopycnals (Figure 14). Nutrient stocks are homogeneous in the deeper parts of the two basins: for isopycnals higher than 29.1, concentrations were about 8.5 μmol/kg in nitrate and 0.38 μmol/kg in phosphate. The light surface layers, lower than 28.1 in the Algerian Basin and lower than 28.55 in the Tyrrhenian

Sea, were depleted in nutrients. Underneath and for both regions, nutrient concentrations increased until isopycnal 28.65, by 3.5 μmol/kg in nitrate and 0.08 μmol/kg in phosphate. Then differences appear between the two profiles. In the Algerian Sea, the nutricline gets sharper between isopycnals 28.65 and 29.05. The base of the nutricline is inside LIW (350-500 dbar), therein nutrient concentrations reach their maximum value. They slightly decrease in the transition zone (500-1500 dbar) towards their DW concentrations, by 0.7 μmol/kg in nitrate and 0.02 μmol/kg in phosphate. In the Tyrrhenian Sea, the extension of the

nutricline is reduced to the isopycnal 29 at 250 dbar: nutrient concentrations reach a local maximum above LIW, then they slightly decrease inside the LIW core (salinity maximum at 400 dbar, Figure 13). A second major difference with respect to the Algerian Basin appears in the transition zone (500-2000 dbar): therein nutrient concentrations sharply increase by 2.3 μmol/kg in nitrate and 0.14 μmol/kg in phosphate, while gradients had opposite sign in the neighboring basin.

The diapycnal diffusive fluxes of nutrients are computed using the parameterizations detailed in Section 2.3. Let us remind

here that two different diapycnal mixing processes are alternatively considered, whether turbulent diffusion through the nutricline or salt fingering across the transition zone. The diffusive fluxes between surface and LIW are computed with respect to the gradient of nutrients across isopycnals and gross estimates of the kinetic energy dissipation rate (Equation 7). The diffusive fluxes between LIW and DW are computed by a diffusion coefficient $K_{sf}$ given by the Radko and Smith (2012) formulation (Equation 5), that only depends on the bulk density ratio (Equation 1). Bulk ratios respectively equal to 1.38 and

1.32 yield $K_{sf} = 3.0 \cdot 10^{-5}$ m²/s in the Algerian Basin and $K_{sf} = 3.3 \cdot 10^{-5}$ m²/s in the Tyrrhenian Sea. The obtained fluxes are displayed over the layout of water masses (Figure 15). It appears that the largest fluxes correspond to the nutrient supply in the surface productive waters. Regarding LIW stocks, there is a loss of nutrients in the Algerian Basin, mostly towards the surface layer, by 34 μmol/m²/d in nitrate and 1.76 μmol/m²/d in phosphate. Whereas in the Tyrrhenian Sea, there is an enrichment by 19 μmol/m²/d in nitrate and 0.34 μmol/m²/d in phosphate. The contribution from the DW reservoir through



thermohaline staircases represents one fourth of that supply; considering that LIW receives also 14 μmol/m²/d from above during Tyrrhenian transit, LIW is a net exporter of nutrients from the basin. The implications will be further discussed in comparison with other processes in Section 4.4.

## 4. Discussion

### 4.1 Different vertical structures

The two staircase regions crossed by the cruise PEACETIME have displayed different layering aspects. In the Tyrrhenian Sea, thermohaline staircases are the predominant feature of CTD profiles (Figure 3); they spread over most of the water column from 600 to 2275 dbar; they are made up of ten mixed layers that can reach atypical thicknesses of 330 m. Only the upper part of the structure has been observed by the BGC-Argo floats because their profiling depth was limited to 1000 dbar. On the other hand, the thermohaline staircases of the Algerian Basin are well documented by the BGC-Argo collection; they extend between

600 and 1200 dbar depth (Figure 7), with seven mixed layers of thickness 30-75 m. How to explain these structural differences between two basins occupied by the same water masses? One reason is the inflow of newly formed DW in the Algerian Basin: denser than the old ones, they lay on the seafloor up to 2300 dbar (Figure 7) uplifting the "older" DW (Zunino et al., 2012; Send and Testor, 2017). Consequently, the DW pool presents a temperature-salinity minimum at 1400 dbar that intrinsically limits the vertical extension of the structure. In the Tyrrhenian Sea, the transition zone is not bound so that staircases can

extend over larger depths. On the other hand, Tyrrhenian LIW are warmer and saltier than Algerian LIW, they favor low density ratios which drive preferential development of large layers (further detailed below in this Section 4.1).

The thermohaline staircases observed during the cruise PEACETIME have similar characteristics than that previously reported in the same basins. Regarding the Tyrrhenian Sea, the vertical structure is very close to those observed in 2007-2010, reported by Falco et al. (2016), and more interestingly, to the one observed 44 years ago, in May 1973, reported by Molcard and Trait

(1977). Meanwhile, Zodiatis and Gasparini (1996) observed a lower number of layers inside the same vertical extension (only 6 in 1991, even 5 in 1992), together with larger individual thicknesses (up to 543 m). Notably, the DW properties were modified during this period as an effect of the eastern Mediterranean transient (Gasparini et al., 2005): the observed injection of heat and salt in the deep Tyrrhenian Sea would have favored the lowest layers to merge. Regarding the Algerian Basin, the present vertical structure is close to the ones observed in 1994, reported by Krahmann (1997). At that time, the vertical structure

extends between 500 and 1300 dbar, with seven layers of thickness ranging between 28 m and 67 m in average over two field surveys in winter and autumn. Considering another report by Bryden et al. (2014) from biannual surveys between 2006 and 2010, the transition zone extended in depth, however limited by a temperature-salinity minimum at 1600 dbar, with eight layers averaging 93 dbar in thickness and a bulk density ratio of 1.28. In comparison with the present observation, an uplift of the "old" DW by 200 m and an increase of the density ratio to 1.38 could explain such structural differences.

As sketched out above, layers tend to be thicker when bulk density ratios decrease. The theoretical model of Radko (2005) describes the macroscopic formation of thermohaline staircases by salt fingering as a series of merging events that make thin





and unsteady layers grow into an equilibrated vertical structure. At this ultimate state, layer thicknesses reach a critical value determined by the bulk density ratio. The model predicts critical heights to drop down with increasing bulk density ratios, about 200 m for bulk density ratios of the Tyrrhenian Sea (1.32) and about 80 m for those of the Algerian Basin (1.38).

According to the present observations (Tables 2 and 3), the critical heights of layers in the middle of the structure are in good agreement with the model predictions. Moreover, following Radko et al. (2014), there is a tendency for unsteady layers to merge if the height of the adjoining layers is lower than their critical value; the preferential merging scenario, denominated "B-merger", should be realized in a manner that temperature-salinity steps between thick layers is maintained. This mechanism of adjustment has been recorded in the present observations. The daily profiles at long stations in the Tyrrhenian Sea (casts 1-

4, Figure 3) and in the Algerian Basin (casts 2-7, Figure 7) documented the occurrence of small transient layers at the interface between layers that have likely reached their critical height, without any change of layer properties. In addition, the two sequences of BGC-Argo profiles in the Algerian Basin (Figures 10 and 11) also documented the occurrence of a small transient layer that whether splits the main step 3/4 when the adjacent layers are large or merges when the adjacent layers shorten, while the interlayer temperature-salinity gradients remain equal.

**4.2 Areal extents delineated by large scale circulation features**

In this section, the spatial extension of the staircase regions is first compared to previous observations, then discussed in the view of underlying circulation features. In the Tyrrhenian Sea, the present analysis suggests a unique structure spreading from the central area until the southwest border (Figure 5). According to Zodiatis and Gasparini (1996) that studied a set of cross-shore transects all around the basin, thermohaline staircases cover large distances from the central part, becoming progressively

weaker and finally disappearing near the borders. The authors reported an extension in the southwestern sector of the basin, documented along two cross-shore transects southeast Sardinia, which is in agreement with the present case (Figure 5, left panel). Sparnocchia et al. (1999) confirmed such extension in the Sardinian Channel. More recently, high-resolution synoptic observations of staircases by seismic data have been reported along three zonal transects north of the 40°N parallel (Buffett et al., 2017). The staircases are depicted as continuous stripes, well-ordered in the central part of the basin, weakening close to

the continental slope, with a remarkable spatial continuity. This beam of independent observations confirms the hypothesis of a unique structure extending over large parts of the Tyrrhenian Sea, with an epicenter located at its deep central area, as proposed by Molcard and Tait (1977).

Such epicentral configuration, with active salt fingering (see Section 4.3) and well-developed staircases that progressively erode nearby, can also be applied to the case of the Algerian Basin. The basin-scale survey by BGC-Argo floats pointed out a

region of systematic staircase detection around (37°45'N, 5°20'E) extending west until the 4°E meridian and south until the 37°20'N parallel. In addition, the zonal transect of the cruise PEACETIME, although of coarse resolution, showed a well-developed vertical structure in this region (cast 8, Figure 7). Thermohaline staircases have been previously observed along the same zonal transect (Bryden et al., 2014). Thanks to biannual surveys at higher spatial resolution, the authors reported the different layering aspects: well-ordered vertical structures around 4°E progressively eroded westwards, and jumbled profiles





at the eastern stations. More recently, glider surveys along the 3°E and 4°E meridians provided high resolution synoptic observations of thermohaline staircases with a remarkable spatial continuity between the different layering aspects (Cotroneo et al., 2019). These reports are in agreement with the regionalization by staircase regimes proposed by Krahmann (1997): a region of inverted layers close to Sardinia where LIW enter the basin and flow northwards along its coast, a band around the first of well-ordered layering, and a region further west with "diffusively reduced" layers.

If DW play a role on the vertical extension (Section 4.1), areal extents of thermohaline staircases appear to be controlled by LIW circulation features. In other terms, locally high LIW properties, originating whether from the southeast or from southwest Sardinian sectors, would drive the same epicentral configuration for the two staircase regions. Warmer and saltier LIW are brought throughout the Strait of Sicily by the eastern Mediterranean outflow that directly sinks inside the Tyrrhenian Sea. Following Sparnocchia et al. (1999), the flow is made up of LIW at 200-800 m depth and transitional DW detected down to

1850 m depth. The authors argued that the mixing by salt fingering process acts when the bottom slope does not influence the vein any more. On the other side of the Sardinian Channel, the LIW vein flowing inside the Algerian Basin is warmer and saltier than the central Algerian waters. The vein circulates northward along the continental slope, delineated by a density front along which interleaving layers are triggered (Krahmann, 1997).

The influence of young LIW intrusions inside the adjoining basins can extend over large distances. In the case of the Tyrrhenian

Sea, the deep circulation is weak enough that young LIW, progressively entrained cyclonically along the continental slope, affect the central basin through lateral intrusions (Zodiatis and Gasparini, 1996). In the case of the Algerian Basin, the eastern Algerian Gyre, a component of the basin-scale barotropic cyclonic circulation (Testor et al., 2005), plays a stabilizing role for interleaving layers to take the form of thermohaline staircases, and extend westwards until the epicentral region. Interesting to note, LIW are observed warmer and saltier in the western part of the gyre (between 4°E and 6°E) even if it is the part of the

gyre farthest from the inflow of young LIW (Mallil et al., 2016). The anomaly is also sketched in the transect PEACETIME: the cast 8 located in the western part of the gyre presents saltier LIW than that of cast 9 located in its eastern part (Figure 7). This patch matches with the epicentral region where staircases become active and well-developed (Figure 6). The origin of this patch can be inferred from specific mesoscale structures: the Sardinian eddies. They are anticyclones with a deep LIW core (600 m depth) that detach from the continental slope and evolve at the periphery of the eastern Algerian gyre (Testor and

Gascard, 2005). As a result, young LIW transported by Sardinian eddies likely end up confined in the western part of the gyre with slightly modified properties. In addition, the float 6901513 has been trapped three times inside Sardinian eddies. This is signed by the high anomalies of LIW properties and the reduction of sensing aperture with respect to their deep core (Figures 8 and 9). Two events of July 2013 and July 2014 were located in the northern part of the western Algerian Gyre (39°N, 5-6°E), one event of March-April 2015 was located along the eastern border of the basin.

**4.3 Temporal continuity and water mass conversion**

The temporal continuity of thermohaline staircases is the most challenging aspect to assess from observations. In the present study, the large timeseries collected by the BGC-Argo floats with a resolution of some days revealed the continuity of the





layering pattern over months and years (Figures 4, 8). As a result, adjustments of the vertical structure, or even their decay, have been documented at short timescales (some days) in response to changes of LIW characteristics (changes of immersion or properties, Figures 5, 9). In the case of the Algerian Basin, these changes have been analyzed under the caveat of rapid float motions as resulting spatial variations may insert distortions (Figure 10, 11). Thanks to the BGC-Argo collection, thermohaline staircases are found long-lived, whether in the Tyrrhenian Sea in agreement with the early estimate of Molcard and Tait (1977), or in the Algerian Basin.

Quasi-permanent staircases are thought to be sites of thermohaline changes among continuous mixed layers, characterized according to two specific regimes and identified by range values of lateral density ratios ($R_L$, Equation 3). In a first regime, thermohaline changes can take the form of intrusions of heat and salt spreading horizontally inside layers. This spatial regime is triggered and sustained by isopycnal stirring from a steady interleaving state (McDougall, 1985; Merryfield, 2000), which would yield to lateral density ratios close to 1 (Schmitt, 1994). Alternatively, thermohaline changes can take the form of slow temporal trends induced by the downward flux of heat and salt. In this temporal regime, the lateral density ratio is identifiable with the convergence flux ratio that relates heat and salt fluxes across salt fingers (Schmitt, 1994). Observations often provide a mixture of the two regimes (Schmitt et al., 1987), nevertheless the relative changes in layer properties can be documented whether spatially with synoptic surveys, or temporally with long term records at fix locations. Lateral density ratios derived from observations can be compared to theoretical models in order to identify whether isopycnal stirring or salt fingering is active. As sketched out in Section 3.2, the present estimations of lateral density ratios can be characteristic of one of the two regimes specifically, depending on the spatial or temporal limitations of the considered distributions.

To the best of our knowledge, in situ measurements of lateral density ratios at inter-annual scales, with a resolution ensuring the continuity of the layers among the vertical structure, are made for the first time available for the western Mediterranean achieved by the present BGC-Argo collection. Lateral density ratios have been evaluated in the Tyrrhenian Sea (Zodiatis and Gasparini, 1996) as well as in the Algerian Basin (Krahmann, 1997) considering synoptic field surveys. Their results are in agreement with the theoretical prediction of the spatial regime with values close to 1. An attempt of evaluation of long-term trends using historical data has been proposed by Zodiatis and Gasparini (1996), but the fluctuations encountered in the vertical structure (detailed in Section 4.1) did not lead to any reliable estimate of lateral density ratio within single and continuous layers. Falco et al. (2016) provided interannual trends of vertical averaged properties, which could neither give access to any estimate of lateral density ratios.

The layer temperature and salinity changes have been analyzed in both spatial and temporal scales using the BGC-Argo collection. Regarding the Tyrrhenian Sea, the two floats drifted slowly in the southwest sector, moving progressively away from the epicenter. An estimate of long-term trends would be affected by such a slow spatial motion, with the challenge to be untangled from spatial variations. Moreover, only the upper vertical structure has been sampled by the floats (until 1000 dbar), more likely affected by adjustments and inversions among layers because the eastern Mediterranean flow is preferentially injected in that depth range (Sparnocchia et al., 1999). As a result, the consequent fluctuations reach the order of instrumental





precision (0.01 in salinity), so the estimation of any lateral density ratio has been discarded and not reported for the Tyrrhenian Sea in the present study.

On the other hand, the more dynamical situation of the Algerian Basin provided extensive and synoptic surveys of the area, with several pathways across the eastern Algerian gyre. In particular, the duration of the BGC-Argo timeseries allowed to

assess variations of layer properties larger than the instrumental precision limit, with about 0.02 in salinity during the four years of acquisition. The collection gives access to the main layers (3 and 4) and the largest steps (larger than instrumental precision) of the vertical structure, which separates the layer properties and reduces the noise in the slope estimates. The resulting distributions lay out in a temperature-salinity diagram as a superimposition of episodes marked by distinct segments with an overall trend, where the two regimes of water mass conversion appear untangled (Figure 12). Lateral density ratios

have been found to 0.91 from distributions limited to short episodes. This is in agreement with theoretical results of thermohaline changes driven by isopycnal intrusions (McDougall, 1985; Merryfield, 2000). When the distribution is limited to the western sector of the eastern Algerian Gyre (epicentral region), the lateral density ratio drops down to values in the range of 0.74-0.83 depending on layers. When the whole dataset is considered, the gross lateral density ratio reaches an average value of 0.72. These estimations can be compared to the theoretical flux ratios, equal to 1 for isopycnal stirring process, that

drop down to 0.5-0.7 when salt fingering is active (Schmitt et al., 1987). Flux ratio predicted by the model of Radko and Smith (2012) is 0.60 using bulk density ratio of the Algerian Basin (1.38). This result consolidates the diagnostic based on layering patterns of active salt fingering in the western sector of the eastern Algerian Gyre.

## 4.4 LIW fertilization in the Tyrrhenian Sea

The nutrient pool below the surface productive waters is interconnected over the whole Mediterranean by the thermohaline

circulation. LIW act as conveyor belt that progressively accumulates nutrients from the eastern to the western basins of the Mediterranean Sea. During the cruise PEACETIME, LIW fertilization has been particularly observed across the Tyrrhenian Sea (Figure 1). As evidenced by a sequence of five stations performed in this area (Figure 13), the eastern Mediterranean LIW flowing inside the Tyrrhenian Sea with low nutrient concentrations are able to reverse and accentuate the vertical gradient of nutrient concentrations with DW stocks which originate mostly from the Algero-Provençal Basin. Therefore, the consequent

upward diffusive flux enriches the LIW with nutrients coming from the Algero-Provençal Basin. The enhancement of vertical diffusivity by thermohaline staircases, permanently covering the central part of the Tyrrhenian Sea, suggest that thermohaline staircases likely enrich LIW also with nutrients of transitional waters of Ionian origin.

Before discussing further implications on nutrient dynamics, we analyze the corollary question of physical processes responsible of these modifications. As a prerequisite, the candidate processes should act at long term (months or even years),

such as vertical diffusion, advection and horizontal diffusion. The depth range concerned by these modifications needs also to be clearly specified. The Ionian vein flowing through the Sicily Strait has a bounded extension: the interface at 200 dbar depth delineates the upper limit with the layer above occupied by the modified Atlantic water outflow; the depth of the sill (480 m) delineates its lower limit. In consequence for the Tyrrhenian waters, the surface layer (above 200 dbar) is not influenced by



the Ionian inflow; the deep layer (below 500 dbar) is quickly and locally affected by the cascading of the dense Ionian waters

that are modified by turbulent mixing with DW until they reach their level of equilibrium (between 500 and 1400 dbar, Sparnocchia et al., 1999). As evidenced in Figure 13, only the layer between 200 and 500 dbar gradually changes its large scale properties and increases its nutrient concentrations over the whole Tyrrhenian Sea.

The vertical fluxes of nutrients crossing this depth range (200-500 dbar) have been evaluated in presence of thermohaline staircases, considering two alternative mixing processes: turbulent diffusion above LIW and salt fingering below LIW. Note

that the diffusivity of the latter process has been estimated by the fine structural characterizations reported from the cruise PEACETIME (Sections 2.3 and 3.3). It is about three times larger than the turbulent diffusivity reported by Ferron et al. (2017) in the depth range of 1000-2000 m, which is in agreement with the theoretical prediction of Hamilton et al. (1989) and the resulting enhancement of vertical fluxes by the presence of thermohaline staircases. As reported in Section 3.3 (Figure 15), the nutrient flux between DW and LIW is opposite to the flux of salt in the Tyrrhenian Sea. That nutrient flux represents one fourth

of the nutrient supply inside LIW, the three other fourths are provided by nutrients coming from above LIW via, mostly, the biological pump. Whether the nitrogen, "new" for the Tyrrhenian Sea, derives from the modified Atlantic waters fertilized in the Algerian basin by the intense mesoscale activity and advected in the basin, or from regional terrestrial and atmospheric inputs, it cannot be untangled with the present data.

Regarding the amplitude of LIW enrichment in the Tyrrhenian Sea, it has been quantified between two stations visited during

the cruise PEACETIME, one located northeastern Sicily and the other southwestern Sardinia (Figure 13). The increase of nutrient concentrations (2 μmol/kg in nitrate within a 100 m thick layer) would be equilibrated by the cumulated vertical diffusive fluxes (19 μmol/m$^2$/d in nitrate) on a duration of 29 years. LIW residence times inside the Tyrrhenian Sea should be lower: the dispersion of the MedArgo floats drifting at 350 m is about some years in this area (Poulain et al., 2007). The contribution of physical processes should be replaced in a larger scope, including the action of the biological pump that

resupplies in nutrients the waters below the surface productive layer through remineralization of organic matter and export, including also the contribution of external sources such as atmospheric deposition. Considering the first mechanism, the upward flux of nitrates to the surface layer estimated in the present study (560 μmol/m$^2$/d, Figure 15) corresponds to 16 gC/m$^2$/y of new production in carbon. This is in agreement with new production values of 10 gC/m$^2$/y calculated from the phosphorus budget in the Tyrrhenian Sea (Béthoux, 1989). Moreover, annual carbon fixation rates, equivalent to total

production, were estimated to 87 gC/m$^2$/y from ocean color imagery in the Tyrrhenian Sea (Morel and André, 1991). The turnover rate of nutrients associated to the present upward flux is equal to 18%, which is in agreement with f-ratio values of oligotrophic areas.

All this depicts a complex scenario. Contrary to what could have been supposed looking at the strong nutrient gradient between LIW and DW at the southeast entrance of the Tyrrhenian Sea, the nutrient stock accumulated by LIW along the path in the

eastern Mediterranean and the Sicily Strait, does not fertilize the basin. Instead the Tyrrhenian primary production, estimated in the order of 87 g/m$^2$/y by Morel and André (1991), is supported either from sources, coastal or atmospheric, within the basin and, possibly, by the inputs from the Algerian Basin. The latter inputs arrive very likely in organic form, since existing data

(e.g., Astraldi et al, 2002) do not show significant inorganic nutrients concentrations in the surface along the Sardinian Channel. These nutrients are utilized within the basin, biologically pumped down into the LIW and exported to the Algero-Provençal

Basin, more than compensating the surface and deep inputs from the latter. Overall the Tyrrhenian Sea acts as a remineralization basin and as a source of nutrients for the Algero-Provençal Basin more than a sink for the eastern originating nutrients. Finally, we should add another process that might contribute to LIW enrichment, the isopycnal diffusion at the 400-500 dbar depth horizon of the LIW. This would be in agreement with the development of thermohaline staircases along the continental slope by lateral intrusions (Zodiatis and Gasparini, 1996; Sparnocchia et al., 1999) and with the contribution of

terrestrial inputs which would be transferred to the intermediate layer by the biological pump along the Tyrrhenian shelves.

## 5. Conclusion

Having examined CTD profiles of BGC-Argo datasets and field surveys, we report long-term observations of thermohaline staircases in two sites of the western Mediterranean: the Tyrrhenian Sea and the Algerian Basin. In association with the unprecedented contribution of profiling floats to explore the structural changes of thermohaline staircases, at high resolution

and during several years, their fine characterization carried out by the cruise PEACETIME can be seen from a different perspective. In the two sites, the staircases develop in the transition zone between LIW and DW, over areal extents organized around epicentral regions. These epicentral regions are located inside large scale circulation features, that stabilize the influence of interleaving layers triggered by saltier LIW inflows. As observed in the Algerian Basin, epicentral regions are sites of active mixing, with changes of seawater properties by about +0.06°C in temperature and +0.02 in salinity during the four years of

observation. In-situ lateral density ratios have been analysed in the view of theoretical predictions to identify and untangle i) salt fingering as driver for water mass conversion, with ii) isopycnal diffusion as spreader of heat and salt from the surrounding sources. These processes, together with the biological pump, contribute to observed spatial pattern of the LIW nutrient stocks. A more detailed picture of the long-term evolution of Tyrrhenian staircase and interplay with nutrients will undoubtedly emerge with in course deployments embarking nitrate sensors and profiling capability extended down to 2000 dbar.

**Author contribution**

VT, LP and FDO contribute to the experimental setup, to the data analysis and to the writing of the manuscript. MRA and EPV contribute to the analysis of nutrient data and to the review of the manuscript.

**Acknowledgments**

This study is a contribution to the PEACETIME project (http://peacetime-project.org), a joint initiative of the MERMEX and
CHARMEX components supported by CNRS-INSU, IFREMER, CEA, and Météo-France as part of the program MISTRALS



coordinated by INSU (doi: 10.17600/17000300). Part of the dataset have been acquired during the PEACETIME oceanographic expedition on board R/V *Pourquoi Pas?* in May-June 2017.

This study is a contribution to the following research projects: NAOS (funded by the Agence Nationale de la Recherche in the frame of the French "Equipement d'Avenir" program, grant ANR J11R107-F), remOcean (funded by the European Research Council, grant 246777), and BGC-Argo France (funded by CNES-TOSCA and LEFE-GMMC).

We thank Sandra Helias Nunige, Joris Guittonneau and Patrick Raimbault for the sampling, analysis and inventory of nutrients during the cruise PEACETIME.

We thank the PIs of MOOSE-GE cruises, Laurent Coppola and Pierre Testor, and the PI of the MEDSEA cruise, Patrizia Ziveri, who allowed the deployment and recovery of the BGC-Argo floats Captains and crew of R/V *Tethys II* (CNRS-INSU), R/V *Pourquoi Pas?* (Ifremer) and R/V *Angeles Alvariño* (IEO) who participated to the deployments of the floats are also thanked.

We thank the International Argo Program and the CORIOLIS operational oceanography center that contribute to make the floats data freely and publicly available.

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





| Float WMO | First profile | Last profile | Algerian Basin | Tyrrhenian Sea |
|---|---|---|---|---|
| 6901513 | 09 May 2013 - 0 | 16 Jun 2015 - 182 | 116 / 183 (63%) | / |
| 6901600 | 23 Aug 2014 - 1 | 23 Dec 2015 - 101 | 92 / 102 (90%) | / |
| 6902732 | 16 Jan 2017 - 41 | 18 Dec 2017 - 91 | 16 / 51 (31%) | / |
| 6901491 | 16 Jun 2013 - 0 | 30 May 2015 - 178 | / | 152 / 179 (85%) |
| 6901769 | 31 May 2015 - 0 | 02 May 2017 - 143 | / | 102 / 144 (71%) |
| TOTAL | 09 May 2013 | 18 Dec 2017 | 224 / 336 (67%) | 254 / 323 (79%) |

**Table 1: Selection of CTD profiles collected by five floats operated in the Algerian Basin and in the Tyrrhenian Sea. Each deployment is labeled by the float Word Meteorological Organization (WMO) number. Date and cycle number of the first and the last profiles of this selection, number of profiles (with staircase detection / total, proportion in %).**




| Layer | Depth interval (dbar) Top | Bottom | Layer Thickness (dbar) | Layer Temperature (°C) | Layer Salinity | Step Thickness (dbar) | Temperature Step (°C) | Salinity Step | Density Ratio |
|---|---|---|---|---|---|---|---|---|---|
| 1 | 614 +/- 3 | 660 +/- 1 | 46 +/- 2 | 13.862 +/- 0.000 | 38.720 +/- 0.000 | | | | |
| | | | | | | 4 +/- 1 | 0.079 +/- 0.000 | 0.017 +/- 0.000 | 1.38 +/- 0.03 |
| 2 | 664 +/- 2 | 728 +/- 3 | 64 +/- 3 | 13.784 +/- 0.001 | 38.703 +/- 0.000 | | | | |
| | | | | | | 14 +/- 4 | 0.134 +/- 0.001 | 0.031 +/- 0.000 | 1.32 +/- 0.03 |
| 3 | 741 +/- 6 | 821 +/- 8 | 80 +/- 9 | 13.650 +/- 0.001 | 38.672 +/- 0.000 | | | | |
| | | | | | | 6 +/- 2 | 0.096 +/- 0.001 | 0.022 +/- 0.000 | 1.31 +/- 0.03 |
| 4 | 827 +/- 8 | 956 +/- 8 | 129 +/- 5 | 13.554 +/- 0.001 | 38.650 +/- 0.000 | | | | |
| | | | | | | 28 +/- 6 | 0.192 +/- 0.001 | 0.047 +/- 0.000 | 1.26 +/- 0.01 |
| 5 | 983 +/- 3 | 1259 +/- 15 | 276 +/- 14 | 13.362 +/- 0.000 | 38.603 +/- 0.000 | | | | |
| | | | | | | 56 +/- 10 | 0.147 +/- 0.001 | 0.038 +/- 0.000 | 1.20 +/- 0.00 |
| 6 | 1314 +/- 7 | 1647 +/- 8 | 332 +/- 1 | 13.216 +/- 0.000 | 38.565 +/- 0.000 | | | | |
| | | | | | | 40 +/- 9 | 0.063 +/- 0.001 | 0.018 +/- 0.000 | 1.14 +/- 0.03 |
| 7 | 1687 +/- 7 | 1871 +/- 5 | 184 +/- 8 | 13.153 +/- 0.000 | 38.547 +/- 0.000 | | | | |
| | | | | | | 78 +/- 8 | 0.054 +/- 0.001 | 0.015 +/- 0.000 | 1.16 +/- 0.03 |
| 8 | 1948 +/- 9 | 2033 +/- 20 | 85 +/- 13 | 13.099 +/- 0.000 | 38.532 +/- 0.000 | | | | |
| | | | | | | 35 +/- 15 | 0.022 +/- 0.000 | 0.006 +/- 0.000 | 1.20 +/- 0.02 |
| 9 | 2067 +/- 8 | 2167 +/- 17 | 100 +/- 14 | 13.077 +/- 0.000 | 38.526 +/- 0.000 | | | | |
| | | | | | | 32 +/- 16 | 0.021 +/- 0.000 | 0.007 +/- 0.000 | 1.19 +/- 0.03 |
| 10 | 2199 +/- 8 | 2275 +/- 18 | 76 +/- 17 | 13.056 +/- 0.000 | 38.519 +/- 0.000 | | | | |

**Table 2: Layer and step properties of the staircase in the Tyrrhenian Sea, extracted from the four casts of the cruise PEACETIME (Figure 4). The layers are selected thicker than 15 dbar, the layer numbering is incremented downwards. Parameters are presented as average value and standard deviation over the four casts.**



| Layer | Depth interval (dbar) | | Layer Thickness (m) | Layer Temperature (°C) | Layer Salinity | Step Thickness (m) | Temperature Step (°C) | Salinity Step | Density Ratio |
|---|---|---|---|---|---|---|---|---|---|
| 1 | 611 +/- 21 | 636 +/- 17 | 25 +/- 8 | 13.249 +/- 0.003 | 38.555 +/- 0.001 | | | | |
| | | | | | | 33 +/- 6 | 0.062 +/- 0.002 | 0.013 +/- 0.001 | 1.42 +/- 0.04 |
| 2 | 669 +/- 18 | 713 +/- 15 | 44 +/- 10 | 13.188 +/- 0.001 | 38.542 +/- 0.000 | | | | |
| | | | | | | 32 +/- 8 | 0.054 +/- 0.002 | 0.012 +/- 0.001 | 1.37 +/- 0.05 |
| 3 | 744 +/- 21 | 807 +/- 13 | 63 +/- 12 | 13.133 +/- 0.002 | 38.530 +/- 0.001 | | | | |
| | | | | | | 48 +/- 13 | 0.072 +/- 0.001 | 0.016 +/- 0.000 | 1.31 +/- 0.02 |
| 4 | 850 +/- 16 | 923 +/- 8 | 74 +/- 15 | 13.061 +/- 0.002 | 38.514 +/- 0.000 | | | | |
| | | | | | | 41 +/- 12 | 0.030 +/- 0.001 | 0.007 +/- 0.001 | 1.23 +/- 0.09 |
| 5 | 965 +/- 12 | 999 +/- 4 | 35 +/- 11 | 13.031 +/- 0.003 | 38.507 +/- 0.001 | | | | |
| | | | | | | 35 +/- 9 | 0.039 +/- 0.001 | 0.009 +/- 0.001 | 1.25 +/- 0.06 |
| 6 | 1034 +/- 9 | 1101 +/- 23 | 67 +/- 25 | 12.992 +/- 0.001 | 38.498 +/- 0.001 | | | | |
| | | | | | | 45 +/- 26 | 0.034 +/- 0.002 | 0.008 +/- 0.001 | 1.24 +/- 0.07 |
| 7 | 1154 +/- 13 | 1193 +/- 30 | 39 +/- 29 | 12.957 +/- 0.003 | 38.489 +/- 0.001 | | | | |

**Table 3: Layer and step properties extracted from the nine casts of the cruise PEACETIME in the Algerian basin (Figure 7). The layers are selected thicker than 15 dbar, the layer numbering is incremented downwards. Parameters are presented as average value**
**and standard deviation over the nine casts.**





| Layer | All datasets May 2013 – Dec 2017 | Inside box (37°20'N-38°N, 4°E-6°E) | Float 6901513 Sep 2013 – Dec 2013 | Float 6901600 Nov 2014 – Apr 2015 |
|---|---|---|---|---|
| 1 | 0.78 | 0.82 | 0.89 | 0.91 |
| 2 | 0.77 | 0.83 | 0.90 | 0.98 |
| 3 | 0.73 | 0.83 | 0.93 | 0.94 |
| 4 | 0.67 | 0.76 | 0.90 | 0.89 |
| 5 | 0.65 | 0.74 | 0.91 | 0.82 |
| AVERAGE | 0.72 | 0.80 | 0.91 | 0.91 |

**Table 4: lateral density ratios ($R_L^i$, Equation 3) defined by the changes of temperature and salinity within each layer (i). The average value over the five considered layers is reported. Ratios computed considering the whole dataset (Figure 12, upper left panel), the selection inside the box (37°20'N-38°N, 4°E-6°E) (Figure 12, upper right panel), and the two episodes (Figure 12, lower panels).**



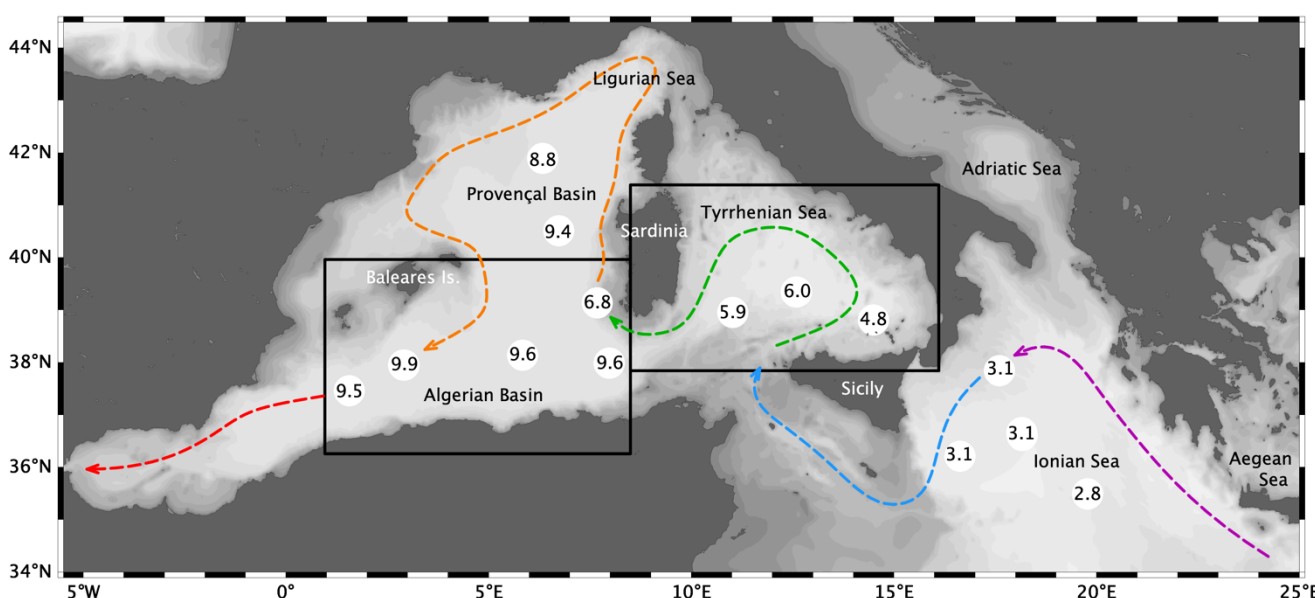

**Figure 1: Indicative pathway of LIW in the western Mediterranean (dash line), crossing the two geographical areas under study (black boxes). Further analyzed in Section 3.3, nitrate concentrations (in µmol/kg) in LIW (immersion of the salinity maximum) as measured during the cruise PEACETIME.**





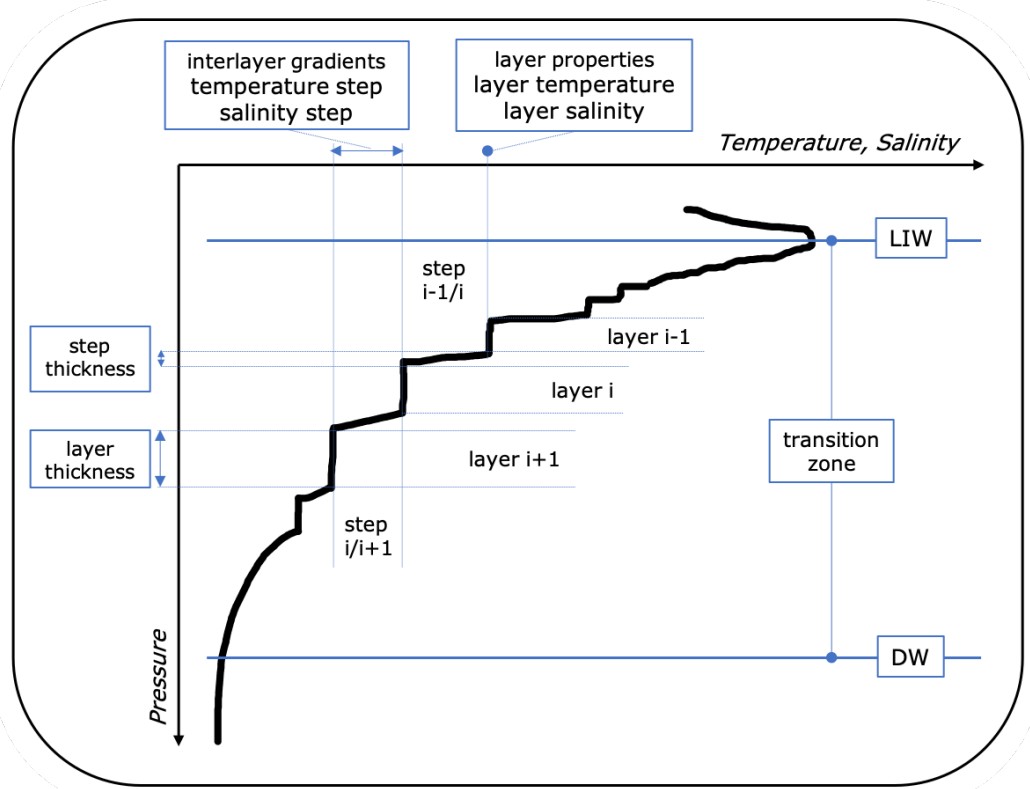

**Figure 2: Drawing of typical thermohaline staircases. Terminology used in the study to characterize the vertical structure. The layer number (i) and the step number (i/i+1) are incremented downwards.**







**Figure 3: Sequence of daily salinity profiles observed in the Tyrrhenian Sea during the cruise PEACETIME. The four casts were performed from the surface to the bottom at the same location. The salinity scale is correct for the profile 1 and each subsequent profile is offset by 0.04.**





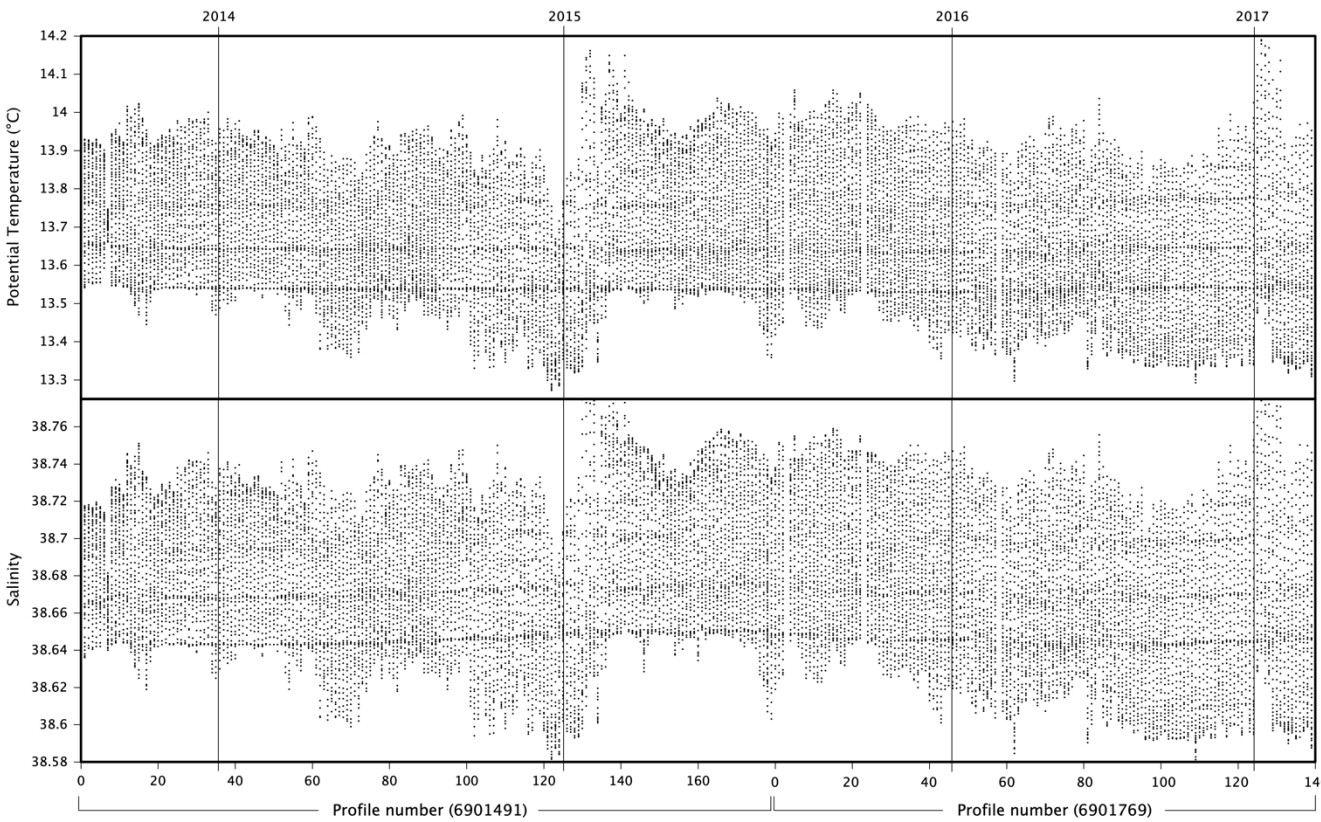

**Figure 4: Temperature (upper panel) and salinity (lower panel) recorded in the range 300-1000 dbar with a vertical resolution of 10 dbar, by the consecutive deployments of the floats 6901491 and 6901769 in the Tyrrhenian Sea. In x-axis, the number of successive profiles with a resolution of one to seven days. The timeframe (in years) is superimposed by vertical lines.**


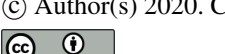



**Figure 5: Left panel, location of staircase detections along the trajectory of the consecutive floats deployed in May 2013 and May 2015. Right panels, time series of layer properties and depth (colored dots) of shipboard and float profiles with staircase detection. LIW properties and depth are indicated in black lines. The 1000 dbar limit is indicated in grey shadows. The layer numbering of the station PEACETIME (Table 2) is indicated inside grey boxes. All panels: station locations and layer properties are indicated in purple for shipboard profiles, in blue for the float 6901491 and in green for the float 6901769.**




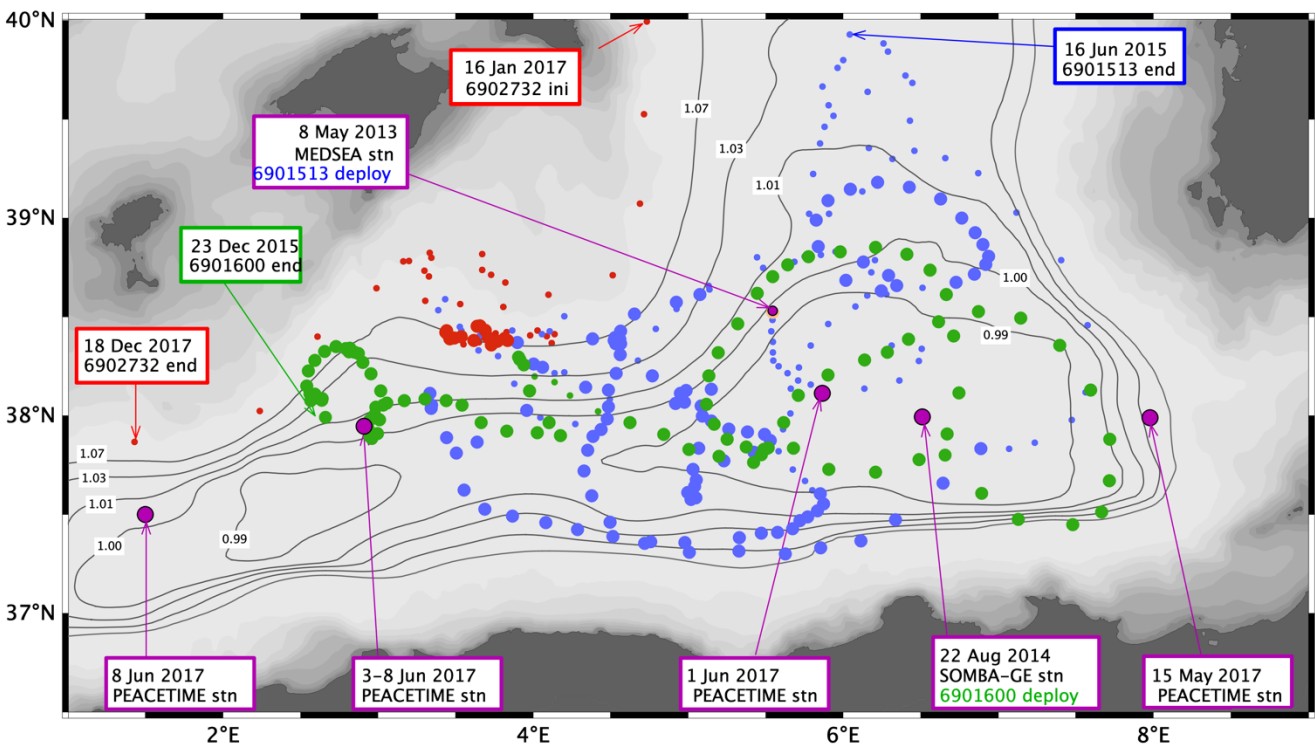

**Figure 6: Locations of the CTD profiles in the Algerian Basin, in large or small dots whether or not staircases were detected. Float collections in blue for 6901513, in green for 6901600, in red for 6902732. Shipboard stations of float deployments and of the cruise PEACETIME in purple. Thin black lines: contours f/H normalized by $f_o$ at 37°45'N and $H_o = 2800$ m.**


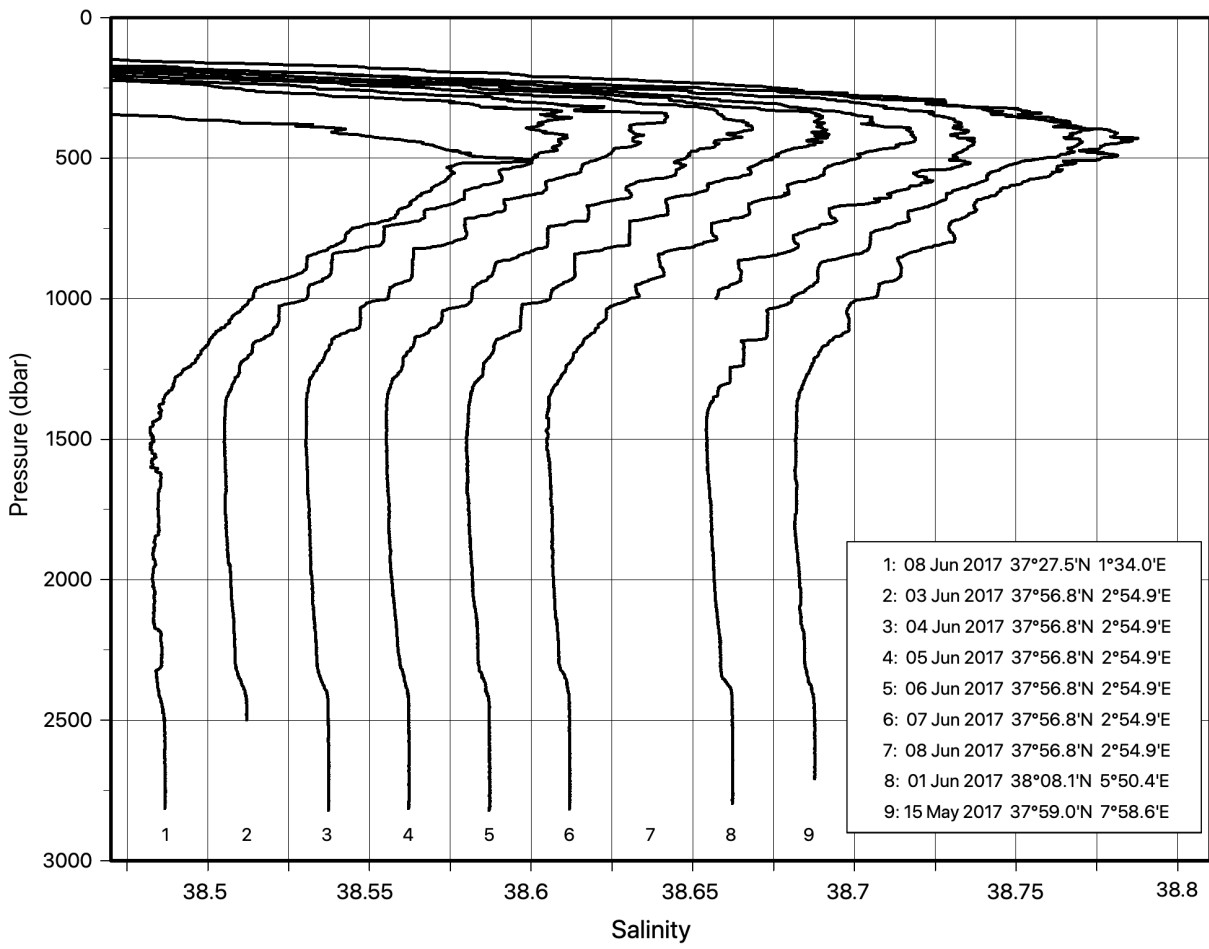

**Figure 7: Sequence of salinity profiles along a zonal transect of four stations across the Algerian Basin during the cruise PEACETIME. The second station was repeated 6 times (casts 2-7). The casts were performed from the surface to the bottom, unless cast 2 down to 2500 dbar and cast 7 down to 1000 dbar. The salinity scale is correct for the profile 1 and each subsequent profile is offset by 0.025.**



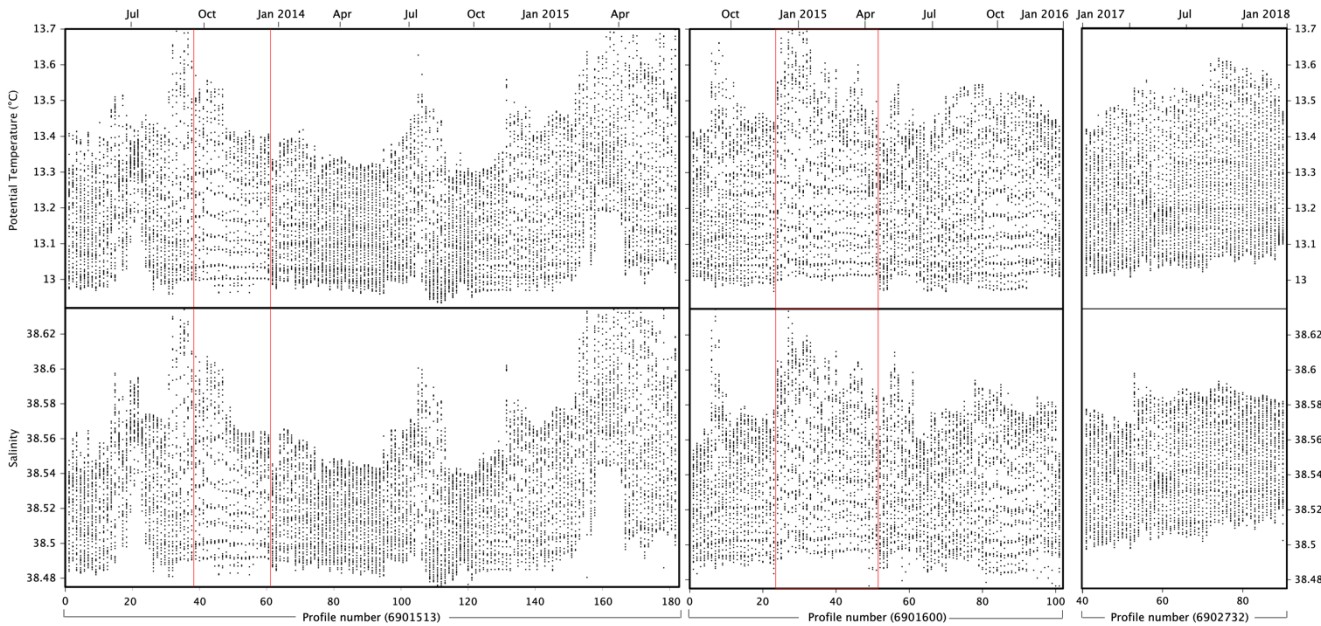

**Figure 8: Temperature (upper panels) and salinity (lower panels) recorded in the range 300-1000 dbar with a vertical resolution of 10 dbar, by the three floats 6901513 (left panels), 6901600 (middle panels) and 6902732 (right panels) deployed in the Algerian Basin. In x-axis, number of the successive profiles with a resolution of one to seven days. The timeframe (in trimesters) is superimposed on top x-axis. The two episodes detailed in Section 3.2 are delimited by red lines.**








**Figure 9:** Time series of layer properties and depth (colored dots) of shipboard and float profiles with staircase detection, for temperature (upper panel) and salinity (lower panel). LIW properties and depth are indicated in black lines. The shipboard CTD profiles are indicated in purple, the float CTD profiles in blue (6901513), green (6901600) and red (6902732). The 1000 dbar limit is indicated by grey shadows. Time series of floats 6901513 and 6901600 are represented in distinct panels to avoid overlapping. The layer numbering of the stations PEACETIME (Table 3) is indicated by grey boxes in the right y-axis. The two episodes detailed in Section 3.2 are delimited by red lines.






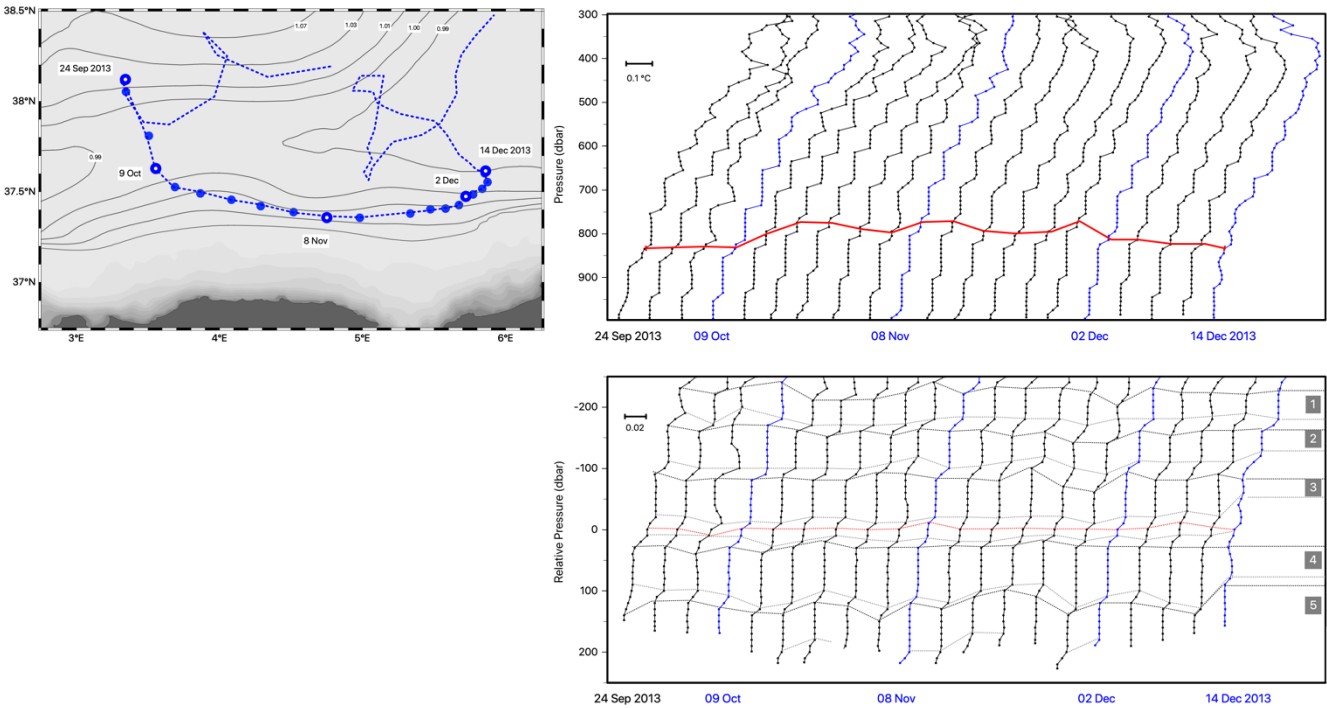

**Figure 10: Successive CTD profiles collected by the float 6901513 between 24 September 2013 and 14 December 2014, with a vertical resolution of 10 dbar. The temperature profiles are shifted by 0.1°C (upper right panel), the salinity profiles are shifted by 0.02**
**(lower right panel). In x-axis, dates of the blue profiles, in correspondence with their location along the float trajectory (empty dots, left panel). In y-axis, pressure referenced to surface (upper right panel), or relative pressure referenced to the depth of the step 3/4 indicated in red lines (lower right panel).**





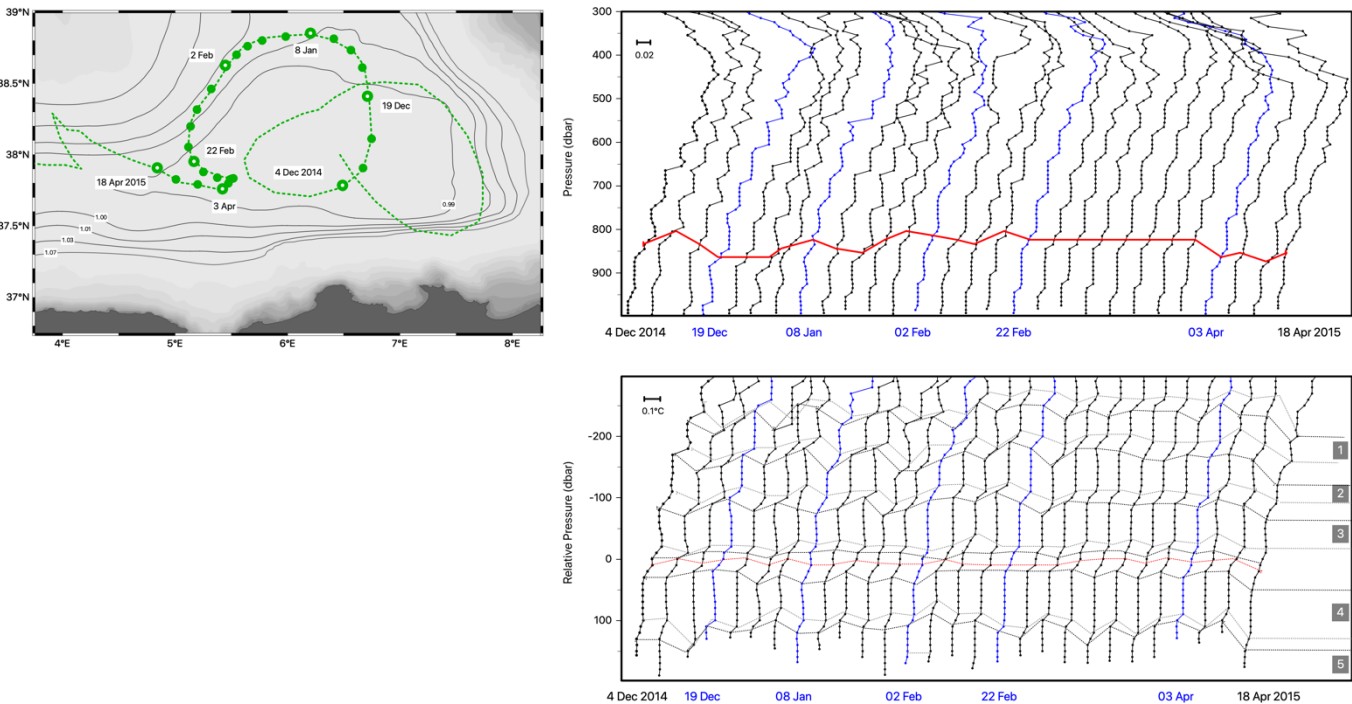

**Figure 11: Successive CTD profiles collected by the float 6901600 between 4 November 2014 and 18 April 2015, with a vertical resolution of 10 dbar. The temperature profiles are shifted by 0.1°C (lower right panel), the salinity profiles are shifted by 0.02 (upper right panel). In x-axis, dates of the blue profiles, in correspondence with their location along the float trajectory (empty dots, left panel). In y-axis, pressure referenced to surface (upper right panel), or relative pressure referenced to the depth of the step 3/4 indicated in red lines (lower right panel).**



**Figure 12: Temperature – salinity diagram of the layer properties. Floats in blue (6901513), green (6901600) and red (6902732); stations PEACETIME (2017) in purple dot, station SOMBA-GE (2014) in purple triangle. The layer numbering is the one of Table 3. The least square fit per layer is indicated in grey lines. Whole dataset (upper left panel), selection inside the box (37°20'N-38°N, 4°E-6°E) (upper right panel), and the two episodes in the lower panels. The least square fit of the upper left panel is reported in the lower panels with dotted lines.**



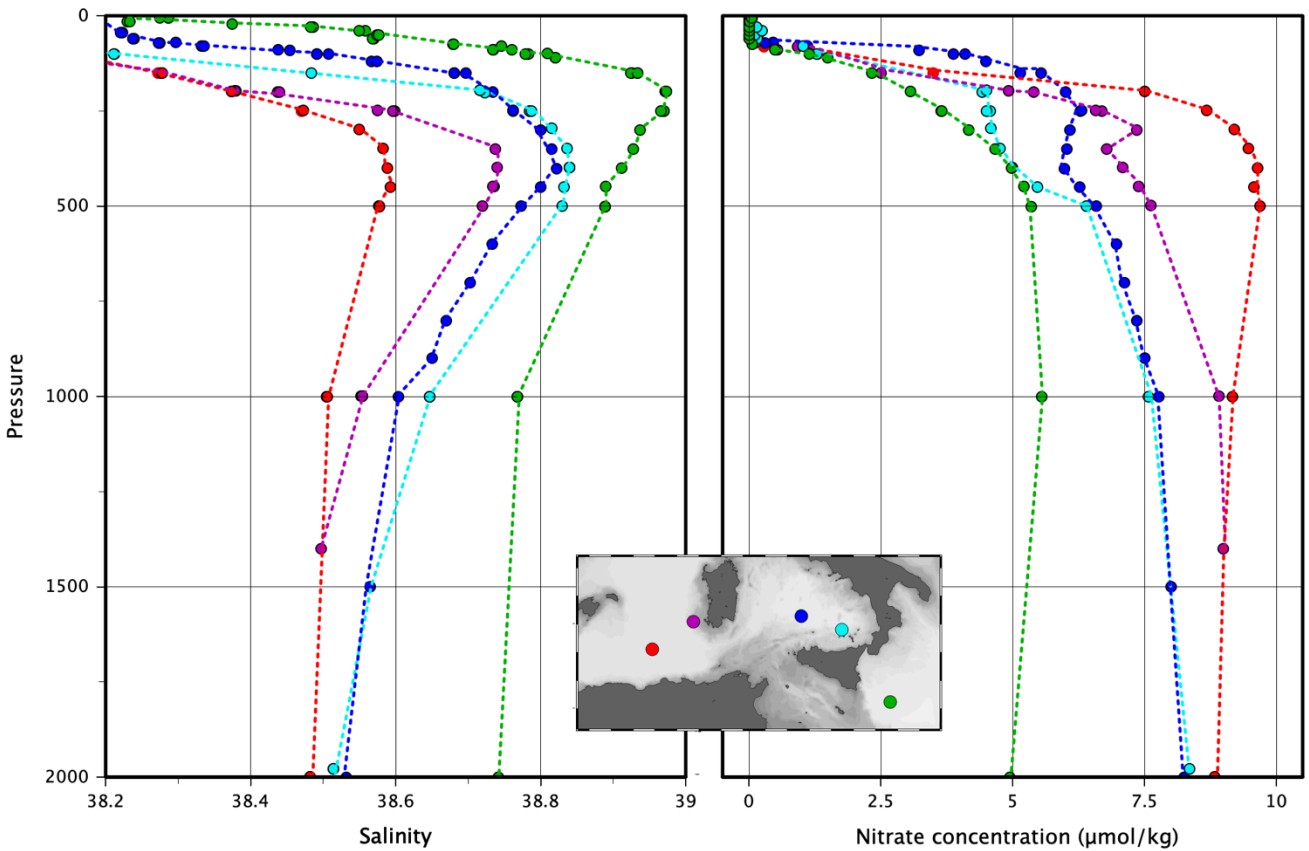

**Figure 13: discrete profiles of salinity and nitrate (in µmol/kg) measured at bottle levels during the cruise PEACETIME. The station in the central Tyrrhenian Sea (dark blue dots) corresponds to the cast 1 reported in Figure 3. The station in the Algerian Basin (red dots) corresponds to the cast 8 reported in Figure 7.**






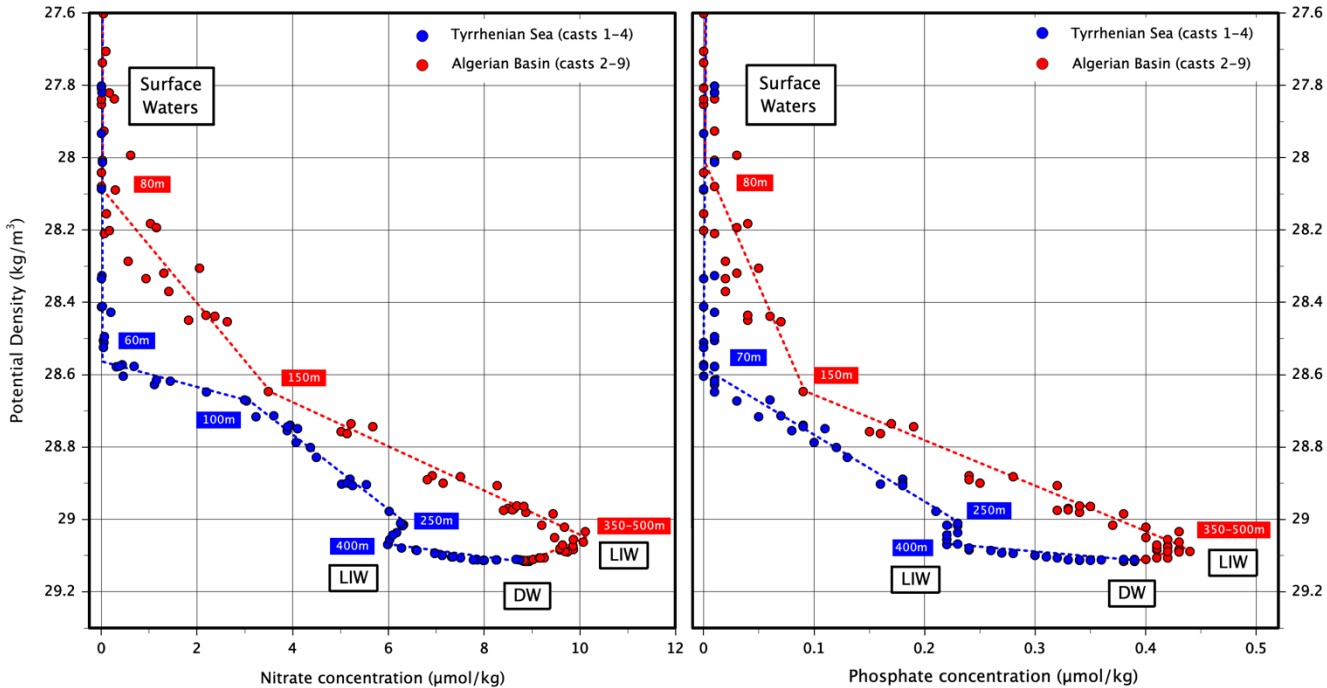

**Figure 14: nitrate and phosphate concentrations (in µmol/kg) against potential density as measured at bottle levels during the cruise PEACETIME. Blue dots: casts 1-4 in the Tyrrhenian Sea (locations indicated in Figure 3). Red dots: casts 2-9 in the Algerian Basin (locations indicated in Figure 7).**





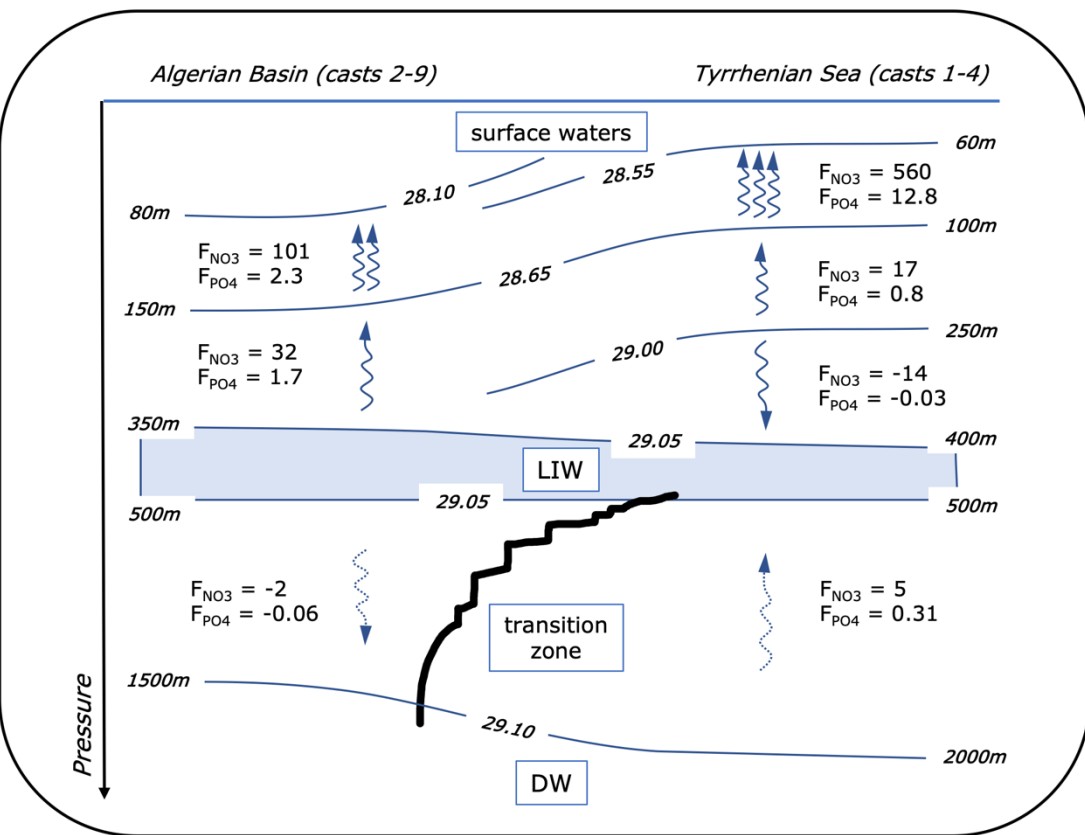

**Figure 15: Vertical fluxes of nitrate $F_{NO3}$ and phosphate $F_{PO4}$ (in µmol/m²/d) in presence of thermohaline staircases. Estimation between surface waters and LIW according to turbulent diffusion (Equation 7), and between LIW and DW according to salt fingering (Equation 5). Positive diffusive flux oriented upwards.**
