# Peer review of "Profiling float observation of thermohaline staircases in the western Mediterranean Sea and impact on nutrient fluxes"

_Biogeosciences, 2019_

## Referee Comment (RC1) · Anonymous Referee #1 · 7 Feb 2020

General comments:

This manuscript presents new measurements of thermohaline staircases from cruises and BGC-ARGO floats in two regions of the western Mediterranea Sea, the Tyrrhenian Sea and the Algerian Basin.

The observation of thermohaline staircases in these regions is not new, and it has already been shown that they develop over epicentral regions confined inside large scale circulation features and are sustained by saltier LIW inflows.

The novelty of this work is in the use, together with data acquired during oceano-graphic cruises, of data from BGC-ARGO floats, which have the potential to sample

an extended areal (following their drift) over a long period of time, which in the specific work is 4.5 years. Although these types of floats include biogeochemical observations, only CTD profiles measured by them are used in the work.

The objective of studying the impact of thermohaline staircases on nutrient fluxes relies only on the nutrient data obtained during oceanographic cruises, one in particular. This raises some doubts about the robustness of the relative calculations, also considering that the part of methods is not very exhaustive on this point.

However, the paper also contains interesting analysis and results, and although it is lacking in some parts, I think it can be considered for publication in this journal after the following issues have been addressed.

Specific comments:

1) Given the preamble in lines 80-85, one expects to find in this work a rich database of nutrients, including from the BGC-ARGO floats. But the analysis of nutrient stocks is only based on 4 stations in the Tyrrhenian Sea and 3 stations in the Algerian basin. I suggest you rewrite this part, giving less emphasis to the biogeochemical observations that are lacking in other studies, and avoiding mentioning that profiling floats can include biogeochemical parameters, because that's not your case.

2) Check the text from line 120 to 125 because there are inconsistencies when compared with Table 1:

line 122: May 2017 should be December 2017 from Table 1, but it is not a date of the cruise PEACETIME;

line 124: float 6901491 was deployed in May 2013 (Figure 5) but the first profile is on 16 June 2013 (Table 1). Why? Is it correct?

3) Terminology (line 141 and following, Figure 2). In this study the vertical region between a mixed layer and the adjacent one in the staircase profile is called "step". Although this definition is found in the literature (for example Bryden et al. 2014), the

term most frequently used to indicate this region is "interface", while the term "steps" generally refers to the overall feature in the profile: "The well defined steps . . . consist of nearly uniform layers separated by thin stratified interfaces" (Radko, J. Fluid Mech., 497, 365-380, 2003). This terminology is also found in Radko, 2005; Zodiatis and Gasparini, 1996; Merryfield, 2000; Falco et al., 2016; Durante et al., 2019; to name a few among many. I therefore suggest using the most common definition, ie the term interface for indicating the vertical region between a mixed layer and the adjacent one in the staircase.

4) Calculation of vertical fluxes (Section 2.3). This is a very important part but some important information is missing or it is not clear enough.

Lines 197 and 201: Why vertical diffusivity "would be" ? What guided the choice of the two methods for calculating it in the two zones? What are the basic assumptions?

Line 195: The statement "the vertical diffusivity of salts (including dissolved inorganic nutrients)" needs a reference. Furthermore, $K\_Sf$ in equation (5) is for Salinity. The salts composing the Salinity contribute in different proportions, and each of them has its molecular diffusivity. Shouldn't you derive the $K\_NO3$ and $K\_PO4$ values from $K\_Sf$ to calculate their fluxes? Or does $K\_Sf$ also apply to these individual components, and if so why? Please explain or add references.

As for equation (6),

- why did you choose 0.2 for mixing efficiency?

- it provides an upper bound for $K\_rho$ (Osborn, 1980). Therefore, also the flux that you calculate with equation (7) will be an upper bound, and you should replace " = " with " $\leq$ " in both equation (6) and (7). I expect this introduces some uncertainty in the consistency of the values shown in Figure 15, with consequences for the conclusions that are drawn from these numbers (section 3.3). Please provide some discussion on the consistency of your calculation.

[Figure]

As for equation (7), Oman and Mahadevan (2015) proposed a model for NO3, based on specific assumptions, that you export to PO4 and to the Mediterranean Sea. Please support it.

5) Line 219: "... Falco et al., 2016", please add Durante et al, 2019.

6) Lines 223 and 279: "which is lower than 1.7, the threshold for the development of thermohaline staircases" ... I suggest to specify "the upper threshold", or to replace with "which is in the range for the development of thermohaline staircases".

7) The paragraph from line 305 to line 308 is quite confusing. Can you perhaps highlight in the Figure 10 the profiles or their parts that you think are "jumbled" by using a different color? Do you really mean that the profiles are "jumbled" or maybe their staircase shape is not well defined in some part of the profile? Also the sentence "the temperature and salinity profiles with depth-decreasing values are locally inverted in the depth range of the transition zone" is not clear. Please rewrite it more clearly. To make it easier to compare upper and lower right panels of Figure 10, can you identify the 5 steps also in the upper panel?

8) Lines 315-320, Figure 11: As previous comment.

9) Lines 411-413: You have not mentioned Durante et al. (2019) who documented enhanced salt finger processes near the bottom after 2010, which are attributed to the ingression of a new denser water mass due to the Western Mediterranean Transition. They show an upward lift of several hundred meters of the steps starting from 2010, and the presence of smaller steps below the deepest thick step, whose number also varies with time, starting from the profile recorded in May 2010 until the end of their series (2016).

10) Line 442: "Sparnocchia et al. (1999) confirmed such extension in the Sardinian Channel". This citation is wrong.

11) Lines 506-508: I agree and I think you should emphasize this new result more in

the Abstract and Conclusion.

Technical corrections:

Line 303: "the float 6901513 drifted westwards" - it is eastwards.

Line 770 (Caption of Table 2): "….(Figure 4)" – it is Figure 3.

―――――――――――――――

---

## Referee Comment (RC2) · Anonymous Referee #2 · 10 Feb 2020

**Review of manuscript bg-2019-504. "Profiling float observation of thermohaline staircases in the western Mediterranean Sea and impact on nutrient fluxes"**

February 10, 2020

**General comments**

In the paper "Profiling float observation of thermohaline staircases in the western Mediterranean Sea and impact on nutrient fluxes" cruise and float temperature and salinity profiles are used to characterize thermohaline staircases in the western Mediterranean. The spatial and temporal coverage of the cruise data is limited, but it is nicely complemented with the float profiles, which show the large spatial extension and temporal persistence of the staircases. The authors also use nutrient profiles collected during the PEACETIME cruise to assess the role of turbulent and salt-finger diffusion for the nutrient enrichment of Levantine Intermediate Waters along their path across the Western Mediterranean basin. My overall evaluation of the manuscript is positive, and I think it should be suitable for publication after some revision.

**Specific comments**

**Manuscript structure.** The goals of the study are quite broad (including a characterization of the structures, its temporal and spatial persistence and their role in the nutrient budgets), and the authors use and mix data from different sources, which makes the manuscript a bit dense sometimes. The novelty of the results should be stressed more clearly from the beginning. For example, I feel the abstract is quite long and contains some general statements, but the description of the main results, their novelty and implications is quite vague (the same applies to the conclusions). I would also suggest to shorten some parts of the manuscript, where many details are given, for example in section 3.2 you could go to more straight the point. That may help to make the manuscript more easy reading. Also, I like that you included phosphorus in the nutrient part, but I don't know how useful it is for the point you want to make, and it increases the manuscript length.

**Nutrient flux calculations and uncertainties.** One of the main novelties of the present study

is to provide estimates of diffusive nutrient fluxes to assess their role for the fertilization of LIW. However, I think the description of the calculation and results are a bit too concise and lack of a serious assessment of the uncertainties (see also next point). For example, for the calculation of the nutrient fluxes it is critical to properly estimate the vertical nutrient gradient (or diapycnal for Equation 7, in the nutricline). Yet, not much information is available about this. Where the gradients calculated from a mean nutrient profile in each basin? How variable are nutrient profiles within a basin (Inter-basin variability seems quite high in Figure 13)? In which depth range was the calculation done and how? The vertical resolution seems quite coarse in the transition layer (Figure 13), how does this affect the results. Overall, uncertainty estimates should be included in Figure 15.

**Uncertainties of diffusivity parameterizations.** The authors should better justify the choice of the diffusivity parameterizations and assess the uncertainties, both for turbulence and double diffusion. Some of the existing parameterizations for salt-fingers diffusion (eg. Kelley, 1990), do not always compare well with direct estimates molecular diffusion across the interfaces (eg. Umlauf et al., 2018). How does the Radko and Smith (2012) formulation compare with the more classical Kelley (1990) parameterization in your case, for example? Regarding turbulent mixing through the nutricline, you used $\varepsilon$ values from the literature. How this affect your flux estimates? What is the magnitude of the uncertainty associated with this assumption? You could consider using some Thorpe-scale based parameterization (eg. Park et al., 2014) applied to the cruise CTD data, to obtain some in-situ estimates of $\varepsilon$. Overturning motions appear evident above the salinity maximum in Figure 7, for example. $\varepsilon$ estimates using this information should be possible. Due to the coarser resolution of the floats, this approach is probably not suitable in this case.

**Representativeness of the nutrient fluxes.** In my view, the strength of the study is the use of float data to significantly extend the spatial and temporal coverage of the observations of thermohaline staircases. On the other hand, the weak point is that this extensive coverage does not apply to the nutrient fluxes. Why you did not use biogeochemical data from the floats? Didn't they include a nitrate sensor? I wonder whether, even if this information is not available, you could still think of using some local potential density – nitrate relationships, or other similar approach, to generalize your results to the float profiles, and better quantify the uncertainties.

**The role of the biological carbon pump for LIW fertilization.** This aspect is briefly discussed in lines 574–582, but I think is relevant. I feel this discussion is a bit insufficient and the mechanism is not well explained, in my view. You suggest that organic matter exported from the photic zone reaches the LIW layer and it is remineralized there, contributing to an important fraction of the observed nutrient enrichment, is that right? It is nice that

you link the nitrate fluxes into the photic zone with other production estimates, but I think you should strength the connection with your observation of the nutrient enrichment of LIW through organic matter remineralization. If this is a dominant mechanism you should observe an increase in apparent oxygen utilization between the Tyrrhenian sea and the Algerian basin in the LIW layer. Do you observe this? Is this comparable to the nutrient increase, in terms of Redfield stoichiometry?

**Technical comments**

**Lines 44–46.** I am not sure whether this sentence is grammatically correct

**Lines 217–220.** Indicate the duration of the station here?

**Line 231.** Maybe "The AMPLITUDE of the temperature-salinity steps.."

**Lines 284 onward.** This part was confusing for me because I had the feeling that the language was guiding me to interpret the observed variations in the layers as temporal variations but the final interpretation seems to be that they are rather a result of spatial inhomogeneities, is that right? Could you stress this a bit at the beginning?

**Line 303.** Did you mean eastwards?

**Lines 305–306.** What is the physical meaning and implications of the fact that the temperature and salinity profiles are inverted within the layers?

**Line 324.** The meaning of this sentence looks not very clear to me: "... confirmed the connection between layers of fluctuating properties, characteristic of spatial variations rather than temporal changes".

**Line 334.** I would say "closer" instead of "close"

**Lines 354–356.** For clarity, you may list the stations in geographical order.

**Lines 384–385.** How do this diffusivities compared with diffusivities through the nitracline?

**Line 404.** "bound" → "bounded", maybe?

**Line 415.** "extends" → "extended"?

**Line 425.** Could you show how you get these numbers?

**Line438.** "According to Zodiatis and Gasparini (1996) that studied", maybe change to "According to Zodiatis and Gasparini (1996), WHO studied".

**Line 449.** "nearby" what?

**Line 482.** "reduction of sensing aperture". Sorry, I am not sure of understanding this. Do you refer to the vertical extent of the layers?

**Figure 4 and 7.** Could you number the layers according to the code in Tables 2 and 3? I think that would help the reader.

**Figure 9.** The red lines delimiting the "events" are thin and difficult to see, could you improve this?

**Figure 10 and 11.** I believe some dates are not correctly reported in the caption.

**Figure 13.** Red and purple dots are not easy to distringuish once printed. Could you maybe use a different color?

**References**

Kelley, D. (1990). Fluxes through diffusive staircases: A new formulation. *Journal of Geophysical Research - Oceans*, 95:3365–3371.

Park, Y. H., Lee, J. H., Durand, I., and Hong, C. S. (2014). Validation of Thorpe-scale-derived vertical diffusivities against microstructure measurements in the Kerguelen region. *Biogeosciences*, 11(23):6927–6937.

Radko, T. and Smith, D. P. (2012). Equilibrium transport in double-diffusive convection. *Journal of Fluid Mechanics*, 692(January 2012):5–27.

Umlauf, L., Holtermann, P. L., Gillner, C. A., Prien, R. D., Merckelbach, L., and Carpenter, J. R. (2018). Diffusive convection under rapidly varying conditions. *Journal of Physical Oceanography*, 48(8):1731–1747.

---

## Author Comment (AC1) · 17 Mar 2020

**BG-2019-504 Responses to referee comments**

Deadline for revision: 19 march 2020

We gratefully thank the reviewers for their thorough reading of the manuscript and their constructive comments. Responses hereinafter and changes in the revised manuscript are indicated in blue.

**Anonymous Referee #1**

General comments:

This manuscript presents new measurements of thermohaline staircases from cruises and BGC-ARGO floats in two regions of the western Mediterranea Sea, the Tyrrhenian Sea and the Algerian Basin.

The observation of thermohaline staircases in these regions is not new, and it has already been shown that they develop over epicentral regions confined inside large scale circulation features and are sustained by saltier LIW inflows.

The novelty of this work is in the use, together with data acquired during oceanographic cruises, of data from BGC-ARGO floats, which have the potential to sample an extended areal (following their drift) over a long period of time, which in the specific work is 4.5 years. Although these types of floats include biogeochemical observations, only CTD profiles measured by them are used in the work.

The objective of studying the impact of thermohaline staircases on nutrient fluxes relies only on the nutrient data obtained during oceanographic cruises, one in particular. This raises some doubts about the robustness of the relative calculations, also considering that the part of methods is not very exhaustive on this point.

However, the paper also contains interesting analysis and results, and although it is lacking in some parts, I think it can be considered for publication in this journal after the following issues have been addressed.

On the revised manuscript,

1. we have added a geochemical dataset collected by BGC-Argo float (nitrate concentrations using SUNA sensor). Data are presented in the dedicated section 2.2, results are presented in the section 3.3, Figures 14-15, and discussed in section 4.4.
2. we have improved the robustness of the estimation of the nitrate fluxes considering different parameterizations of the salt diffusivities, and extending the estimations to the nitrate dataset of the float. The formulations and associated assumptions are presented in section 2.4, the results presented in section 3.3, and Tables 2-6.
3. the original result on the evolution of TS properties inside layers is stressed in conclusion and abstract.

Specific comments:

1) Given the preamble in lines 80-85, one expects to find in this work a rich database of nutrients, including from the BGC-ARGO floats. But the analysis of nutrient stocks is only based on 4 stations in the Tyrrhenian Sea and 3 stations in the Algerian basin. I suggest you

rewrite this part, giving less emphasis to the biogeochemical observations that are lacking in other studies, and avoiding mentioning that profiling floats can include biogeochemical parameters, because that's not your case.

The revised manuscript now includes the analysis of an extended dataset of nitrate concentrations, provided both by the cruise and by one BGC-Argo float. A new section 2.2 has been added to describe this dataset.

2) Check the text from line 120 to 125 because there are inconsistencies when compared with Table 1:

line 122: May 2017 should be December 2017 from Table 1, but it is not a date of the cruise PEACETIME;

This point has been clarified in the revised manuscript.

line 124: float 6901491 was deployed in May 2013 (Figure 5) but the first profile is on 16 June 2013 (Table 1). Why? Is it correct?

Thanks to the reviewer's comment, we realized that they was a mistake in the cruise of deployment: the float 6901491 has been deployed the 16 June 2013 during the cruise VENUS2 instead of the cruise MEDSEA (that performed one station the 27 May 2013 in the vicinity of the location without float deployment). This mistake has been corrected in the text, in Figure 5a and caption of Figure 5. PI and crew members of the VENUS2 cruise have also been acknowledged.

3) Terminology (line 141 and following, Figure 2). In this study the vertical region between a mixed layer and the adjacent one in the staircase profile is called "step". Although this definition is found in the literature (for example Bryden et al. 2014), the term most frequently used to indicate this region is "interface", while the term "steps" generally refers to the overall feature in the profile: "The well defined steps . . . consist of nearly uniform layers separated by thin stratified interfaces" (Radko, J. Fluid Mech., 497, 365-380, 2003). This terminology is also found in Radko, 2005; Zodiatis and Gasparini, 1996; Merryfield, 2000; Falco et al., 2016; Durante et al., 2019; to name a few among many. I therefore suggest using the most common definition, ie the term interface for indicating the vertical region between a mixed layer and the adjacent one in the staircase.

The reviewer's terminology has been followed: "step" changed for "interface", in the text, in Figure 2, and Tables 3, 4 and captions.

4) Calculation of vertical fluxes (Section 2.3). This is a very important part but some important information is missing or it is not clear enough.

Lines 197 and 201: Why vertical diffusivity "would be" ? What guided the choice of the two methods for calculating it in the two zones? What are the basic assumptions?

This section 2.4 has been rewritten following the suggestions of the reviewer: the alternative formulations have been clarified and detailed, the underlying assumptions posed and checked.

Line 195: The statement "the vertical diffusivity of salts (including dissolved inorganic nutrients)" needs a reference. Furthermore, $K\_Sf$ in equation (5) is for Salinity. The salts composing the Salinity contribute in different proportions, and each of them has its molecular diffusivity. Shouldn't you derive the $K\_NO3$ and $K\_PO4$ values from $K\_Sf$ to calculate their

fluxes? Or does K_Sf also apply to these individual components, and if so why? Please explain or add references.

As suggested by the reviewer #2 and in consistency with float dataset, only nitrates are considered in the study (phosphates have been removed). A reference (Hamilton et al. 1989) has been added concerning the equal diffusivity for nitrate and salinity.

As for equation (6),

- why did you choose 0.2 for mixing efficiency?

We choose this value because the studied regime is transitional between the molecular diffusion regime and the energetic regime, under the assumption that the Reynold number is bounded between 8.5 and 400. This point has been clarified section 2.4.

- it provides an upper bound for K_rho (Osborn, 1980). Therefore, also the flux that you calculate with equation (7) will be an upper bound, and you should replace " = " with " ≤ " in both equation (6) and (7). I expect this introduces some uncertainty in the consistency of the values shown in Figure 15, with consequences for the conclusions that are drawn from these numbers (section 3.3). Please provide some discussion on the consistency of your calculation.

The assumption on Reynolds fluxes has been checked in Section 2.4. Following also the comment of reviewer #2, the evaluation of uncertainties in the formulation of diffusivities has been considered in section 2.4.

As for equation (7), Oman and Mahadevan (2015) proposed a model for NO3, based on specific assumptions, that you export to PO4 and to the Mediterranean Sea. Please support it.

As phosphates have been withdrawn from the revised version, this comment has not been addressed.

5) Line 219: ". . . Falco et al., 2016", please add Durante et al, 2019.

This reference has been added in the revised manuscript.

6) Lines 223 and 279: "which is lower than 1.7, the threshold for the development of thermohaline staircases" . . . I suggest to specify "the upper threshold", or to replace with "which is in the range for the development of thermohaline staircases".

This point has been clarified with "upper threshold".

7) The paragraph from line 305 to line 308 is quite confusing. Can you perhaps highlight in the Figure 10 the profiles or their parts that you think are "jumbled" by using a different color? Do you really mean that the profiles are "jumbled" or maybe their staircase shape is not well defined in some part of the profile? Also the sentence "the temperature and salinity profiles with depth-decreasing values are locally inverted in the depth range of the transition zone" is not clear. Please rewrite it more clearly. To make it easier to compare upper and lower right panels of Figure 10, can you identify the 5 steps also in the upper panel?

The proposed modifications on the Figures 10 and 11 suggested by the reviewer have not been followed. It is difficult to identify the layer number in the series with pressure because layer pressures encounter important fluctuations (see the Figure 3 upper right panel), which motivated to change pressure for relative pressure in the lower panel. In the same way, we preferred to describe the "jumbled" profiles from high resolution CTD casts shown in Figure

7. Changes in the text have been done following the suggestion of the reviewer 1 and in agreement to make the section 3.2 lighter asked by reviewer 2.

8) Lines 315-320, Figure 11: As previous comment.

Same as response to the previous comment 7.

9) Lines 411-413: You have not mentioned Durante et al. (2019) who documented enhanced salt finger processes near the bottom after 2010, which are attributed to the ingression of a new denser water mass due to the Western Mediterranean Transition. They show an upward lift of several hundred meters of the steps starting from 2010, and the presence of smaller steps below the deepest thick step, whose number also varies with time, starting from the profile recorded in May 2010 until the end of their series (2016).

The comment of the reviewer has been added to the revised manuscript.

10) Line 442: "Sparnocchia et al. (1999) confirmed such extension in the Sardinian Channel". This citation is wrong.

The geographical extension reported by Sparnocchia et al. (1999) has been corrected with the northwest of Sicily (their section VI and profiles Figure 3a).

11) Lines 506-508: I agree and I think you should emphasize this new result more in the Abstract and Conclusion.

This part is now mentioned in the abstract and conclusion of the revised manuscript.

Technical corrections:

Line 303: "the float 6901513 drifted westwards" - it is eastwards.

Done

Line 770 (Caption of Table 2): ". . ..(Figure 4)" – it is Figure 3.

Done

---

## Author Comment (AC2) · 17 Mar 2020

**BG-2019-504 Responses to referee comments**

Deadline for revision: 19 march 2020

We gratefully thank the reviewers for their thorough reading of the manuscript and their constructive comments. Responses hereinafter and changes in the revised manuscript are indicated in blue.

**Anonymous Referee #2**

General comments

In the paper "Profiling float observation of thermohaline staircases in the western Mediterranean Sea and impact on nutrient fluxes" cruise and float temperature and salinity profiles are used to characterize thermohaline staircases in the western Mediterranean. The spatial and temporal coverage of the cruise data is limited, but it is nicely complemented with the float profiles, which show the large spatial extension and temporal persistence of the staircases. The authors also use nutrient profiles collected during the PEACETIME cruise to assess the role of turbulent and salt-finger diffusion for the nutrient enrichment of Levantine Intermediate Waters along their path across the Western Mediterranean basin. My overall evaluation of the manuscript is positive, and I think it should be suitable for publication after some revision.

We have addressed all the comments proposed by the reviewer, details are given in the responses of each comment.

Specific comments

**Manuscript structure.** The goals of the study are quite broad (including a characterization of the structures, its temporal and spatial persistence and their role in the nutrient budgets), and the authors use and mix data from different sources, which makes the manuscript a bit dense sometimes. The novelty of the results should be stressed more clearly from the beginning. For example, I feel the abstract is quite long and contains some general statements, but the description of the main results, their novelty and implications is quite vague (the same applies to the conclusions). I would also suggest to shorten some parts of the manuscript, where many details are given, for example in section 3.2 you could go to more straight the point. That may help to make the manuscript more easy reading. Also, I like that you included phosphorus in the nutrient part, but I don't know how useful it is for the point you want to make, and it increases the manuscript length.

Following the Reviewer's suggestion, several parts of the manuscript have been shortened, in particular the section 3.2, and the phosphate profiles and fluxes have been removed. The novelty of the results has been clarified in the abstract and the conclusion of the revised manuscript.

**Nutrient flux calculations and uncertainties.** One of the main novelties of the present study is to provide estimates of diffusive nutrient fluxes to assess their role for the fertilization of LIW. However, I think the description of the calculation and results are a bit too concise and lack of a serious assessment of the uncertainties (see also next point). For example, for the calculation of the nutrient fluxes it is critical to properly estimate the vertical nutrient gradient

(or diapycnal for Equation 7, in the nutricline). Yet, not much information is available about this. Where the gradients calculated from a mean nutrient profile in each basin? How variable are nutrient profiles within a basin (Inter-basin variability seems quite high in Figure 13)? In which depth range was the calculation done and how? The vertical resolution seems quite coarse in the transition layer (Figure 13), how does this affect the results. Overall, uncertainty estimates should be included in Figure 15.

A particular attention has been drawn to the description of the nitrate dataset (new section 2.2) and the calculation of the nitrate fluxes (now detailed in table 6, with the identification of depth/density/nitrate intervals from the Figures 13, 14). The flux calculations have been reconsidered with increasing uncertainties over three distinct cases (last paragraph of the section 3.3). The Figure 15 now specifies a range of values for nitrate fluxes.

**Uncertainties of diffusivity parameterizations.** The authors should better justify the choice of the diffusivity parameterizations and assess the uncertainties, both for turbulence and double diffusion. Some of the existing parameterizations for salt-fingers diffusion (eg. Kelley, 1990), do not always compare well with direct estimates molecular diffusion across the interfaces (eg. Umlauf et al., 2018). How does the Radko and Smith (2012) formulation compare with the more classical Kelley (1990) parameterization in your case, for example? Regarding turbulent mixing through the nutricline, you used ε values from the literature. How this affect your flux estimates? What is the magnitude of the uncertainty associated with this assumption? You could consider using some Thorpe-scale based parameterization (eg. Park et al., 2014) applied to the cruise CTD data, to obtain some in-situ estimates of ε. Overturning motions appear evident above the salinity maximum in Figure 7, for example. ε estimates using this information should be possible. Due to the coarser resolution of the floats, this approach is probably not suitable in this case.

Following the reviewer's suggestion, two other formulations have been considered to compute the salt diffusivities across the transition zone (detailed in section 2.4). They provide a range of uncertainties for this parameter, that have been considered in the evaluation of nitrate fluxes (Table 6). Note that the formulation of Kelley (1990) is only applicable to diffusive convection (instable temperature profiles). In the same way, the Thorpe-scale based parameterization suggested by the reviewer has been followed to build a look-up table for dissipation rates (presented in table 2). The values of the look-up table have been discussed with historical measurements (section 2.4) and used to provide a range of values for nitrate fluxes (table 6 and Figure 15).

**Representativeness of the nutrient fluxes.** In my view, the strength of the study is the use of float data to significantly extend the spatial and temporal coverage of the observations of thermohaline staircases. On the other hand, the weak point is that this extensive coverage does not apply to the nutrient fluxes. Why you did not use biogeochemical data from the floats? Didn't they include a nitrate sensor? I wonder whether, even if this information is not available, you could still think of using some local potential density – nitrate relationships, or other similar approach, to generalize your results to the float profiles, and better quantify the uncertainties.

Following the reviewer's suggestion, SUNA data collected by the float 6901769 deployed in the Tyrrhenian Sea (the only one equipped among the 5 considered floats) has been added. This dataset is described in section 2.2. It provides a mapping of the particular nitrate layout of the local nitrate maximum above LIW inferred by nutrient-rich waters coming from the

Algerian Basin (new paragraph in section 3.4). It also allows to quantify the uncertainty of the nitrate fluxes above LIW, over a broader range of situations, seasons and areas (new paragraph in section 3.4 and Figure 15).

**The role of the biological carbon pump for LIW fertilization.** This aspect is briefly discussed in lines 574–582, but I think is relevant. I feel this discussion is a bit insufficient and the mechanism is not well explained, in my view. You suggest that organic matter exported from the photic zone reaches the LIW layer and it is remineralized there, contributing to an important fraction of the observed nutrient enrichment, is that right? It is nice that you link the nitrate fluxes into the photic zone with other production estimates, but I think you should strength the connection with your observation of the nutrient enrichment of LIW through organic matter remineralization. If this is a dominant mechanism you should observe an increase in apparent oxygen utilization between the Tyrrhenian sea and the Algerian basin in the LIW layer. Do you observe this? Is this comparable to the nutrient increase, in terms of Redfield stoichiometry?

The Section 4.4 has been re-organized and rewritten to clarify the proposed mechanisms of LIW fertilization in the Tyrrhenian Sea. The suggestion of the reviewer to explore further the mechanism of remineralization using apparent oxygen utilization has not been inserted to the revised manuscript because the addition of new parameters, the presentation of new notions would extend and heavy the manuscript, compared to the added value of the result. This point will certainly be developed using in course deployments of BGC-Argo floats embarking nitrate and oxygen sensors.

[Figure]

The Redfield ratio AOU / NO3 is equal to +10 in theory. It is measured close to 10 in Algerian waters between the entrance of the Ionian inflow and the central station, as well as in LIW between the central station and the Tyrrhenian outflow.

Technical comments

Lines 44–46. I am not sure whether this sentence is grammatically correct

The sentence has been rephrased in the revised version.

Lines 217–220. Indicate the duration of the station here?

The six days duration is now indicated.

Line 231. Maybe "The AMPLITUDE of the temperature-salinity steps.."

Done

Lines 284 onward. This part was confusing for me because I had the feeling that the language was guiding me to interpret the observed variations in the layers as temporal variations but the final interpretation seems to be that they are rather a result of spatial inhomogeneities, is that right? Could you stress this a bit at the beginning?

This part has been shortened considering the reviewer's comment and the sentence "The two episodes are further analyzed in their geographical context" has been added before the description of the two episodes.

Line 303. Did you mean eastwards?

Yes, it is corrected in the revised version.

Lines 305–306. What is the physical meaning and implications of the fact that the temperature and salinity profiles are inverted within the layers?

This observation reveals lateral intrusions of heat and salt flowing inside the structure along isopycnals. A sentence has been added to the revised manuscript.

Line 324. The meaning of this sentence looks not very clear to me: "... confirmed the connection between layers of fluctuating properties, characteristic of spatial variations rather than temporal changes".

This sentence has been rephrased accordingly, stressing the continuous characterizations of the layers rather than the changes of layer properties.

Line 334. I would say "closer" instead of "close"

Done

Lines 354–356. For clarity, you may list the stations in geographical order.

The list follows the LIW pathway in the revised manuscript.

Lines 384–385. How do this diffusivities compared with diffusivities through the nitracline?

The diffusivities across the nitracline are now estimated using Thorpe-scale parameterization. They are reported in the table 2. Note that the comparison between Table 6 and Table 2 gives the same order of magnitude (not mentioned in the revised manuscript).

Line 404. "bound" → "bounded", maybe?

This change has been done.

Line 415. "extends" → "extended"?

This change has been done.

Line 425. Could you show how you get these numbers?

The critical heights as predicted from bulk density ratios are shown in the Figure 3 of Radko (2005). This is clarified in the revised manuscript.

Line438. "According to Zodiatis and Gasparini (1996) that studied", maybe change to "According to Zodiatis and Gasparini (1996), WHO studied".

This change has been done.

Line 449. "nearby" what?

This sentence has been clarified.

Line 482. "reduction of sensing aperture". Sorry, I am not sure of understanding this. Do you refer to the vertical extent of the layers?

Right, this part was not clear: the usual ranges of temperature and salinity are truncated inside the Sardinia eddies as they are considered on fixed depth ranges. Because these structures have a deeper LIW core, profiles are observed shifted downwards and "cut" at the 1000m profiling depth. This point has been clarified in the revised manuscript.

Figure 4 and 7. Could you number the layers according to the code in Tables 2 and 3? I think that would help the reader.

Layers numbers have been added in Figure 3 and 7, and the cast numbers have been changed by capital letters to avoid confusion.

Figure 9. The red lines delimiting the "events" are thin and difficult to see, could you improve this?

The red lines are thicker in the new Figure 9.

Figure 10 and 11. I believe some dates are not correctly reported in the caption.

The dates in the captions have been corrected.

Figure 13. Red and purple dots are not easy to distringuish once printed. Could you maybe use a different color?

The purple color has been changes for brown color in the new Figure 13.

---

## Author Response (AR1)

**BG-2019-504 Responses to Editors' comments**

Deadline for revision: 16 april 2020

**Co-Editor-in-Chief**

The manuscript could be significantly improved and shortened by checking the language. I have taken the liberty of making suggestions on the attached annotated manuscript up to line 219 (beginning of section 3. Results) and would urge the authors to have the final version checked by an English native speaker.

Thank you very much for your thorough reading and comments on the manuscript. They have been taken into account in the present revised version, and "propagated" after the section 3. Unfortunately, we did not have the possibility to go further given the short deadline and the particular working dispositions that we all encounter in these days. However, in response the comment of the reviewer 2, several parts of the manuscript have been shortened and rephrased during the previous round.

**Associate Editor**

The abstract still needs some work to properly highlight "the main results, their novelty and implications" as requested rightly by reviewer 2. Indeed, in its present form, the abstract, very similar to the previous version does not present clearly enough what is highlighted in the work presented: what are the main messages?

The abstract has been rewritten in order to highlight the main messages (L.12-28).

There is one important point raised by rev2 about the role of the biological carbon pump for LIW fertilization. Although you provided new interesting figures to explore the mechanism of remineralization using AOU, you are defending your choice not to develop that aspect in the revised ms. I agree that this would probably extend too much the text that is already quite dense, and this is not central for the present work. But considering the work that you have done to answer to this comment, I recommend that you present more explicitly than is currently the case, this interesting follow up, at the end of the conclusion.

A sentence has been added at the end of the conclusions (L.645-648).

Nitrate dataset: need to mention (section 2.2) that there are 2 sets of analysis and that this study is focusing on high concentrations below the SML (just to be consistent with the data set presented in the introduction paper).

Done (L.136-139)

L23: remove "s" at "floats" "Nitrate profiles collected by the floats" (only one float with SUNA data)

This sentence has been removed from the abstract.

L243 replace "by the way" by "it has to be noted"

Done

---

## Author Response (AR2)

**BG-2019-504 Response to Editor's comments**

Thank you very much for your thorough reading and comments on the manuscript. They have been taken into account in the present revised version.

---

## Editor Decision (ED2)

**Profiling float observation of thermohaline staircases in the western Mediterranean Sea and impact on nutrient fluxes**

Vincent Taillandier[1], Louis Prieur[1], Fabrizio D'Ortenzio[1], Maurizio Ribera d'Alcalà[2,3], Elvira Pulido-Villena[4]

[1] CNRS, Sorbonne Universités, Laboratoire d'Océanographie de Villefranche, UMR7093, Villefranche-sur-Mer, France
[2] Department of Integrative Marine Ecology, Stazione Zoologica Anton Dohrn, Napoli, Italy
[3] Istituto per lo Studio degli Impatti Antropici e Sostenibilità in Ambiente Marino, CNR, Roma, Italy
[4] Aix-Marseille Université, CNRS, Université de Toulon, IRD, Mediterranean Institute of Oceanography, UMR7294, Marseille, France

*Correspondence to*: Vincent Taillandier (taillandier@obs-vlfr.fr)

**Abstract.** In the western Mediterranean Sea, Levantine intermediate waters (LIW), that circulate below the surface productive zone, progressively accumulate nutrients along their pathway from the Tyrrhenian Sea to the Algerian Basin. This study addresses the role played by diffusion in the enrichment of LIW , a process particularly  inside step-layer structures  down to deep waters : the thermohaline staircases.  Float observations confirmed that  structures develop over epicentral regions confined inside large scale circulation features and maintained by saltier LIW inflows on the periphery. Thanks to  sampling  a four-years period 2013-2017, float observations reveal the temporal continuity of the layering pattern encountered during the cruise, and,  the evolution of  layer properties by about +0.06°C in temperature and +0.02 in salinity. The analysis of in-situ lateral density ratios  conducted  untangle  double-diffusive convection as driver of thermohaline changes inside epicentral regions,  ii) isopycnal diffusion  heat and salt from the surrounding sources. In the Tyrrhenian Sea, the nitrate flux across thermohaline staircases is opposite to the downward salt flux . It contributes  the total  by vertical transfer.  the enrichment of LIW  is  more by other sources, coastal or atmospheric,  
[revised manuscript text omitted]

During  second episode  four months,  float 6901600 completed a  profiling every 10 km along  path of 60 km radius inside the eastern Algerian Gyre (Figure 11, left panel). Lateral intrusions  in most of the area  by the float (until 22 February 2015, Figure 11 upper right panel),  the sector north of  38°10'N  and east of the 5°40'E . In contrast, the layering is  profiles collected in the neighborhood of (37°45'N, 5°20'E). As , local inversions within layers are associated to changes  layer properties (Figure 9, right panel between red lines).  with  distance to the location (37°45'N, 5°20'E) documents the development of lateral intrusions  the surrounding.

The two episodes  the spatial extension sketched out in Figure 6, with active well-ordered thermohalines staircases confined inside the eastern Algerian Gyre, and their progressive erosion all around. Moreover, these episodes confirmed the connection between layers of fluctuating properties and their continuity over the whole period of observation (suggested in Figures 8 and 9).  layer properties in a temperature-salinity diagram,  are aggregated by layer along separated lines (Figure 12, upper left panel). The float and the cruise records are distributed from the oldest to the newest along these lines, with  succession  float 6901513 (blue), SOMBA-GE (purple triangles), float 6901600 (green), PEACETIME (purple dots), and float 6902732 (red). As a result, these lines document  inter-annual  as the five connected layers get warmer by about 0.06°C and saltier by about 0.02 during the four years of observation.

The changes  layer properties  in terms of lateral density ratio expressed in Eq. (3) and estimated from the slope of layer distributions in the temperature-salinity diagrams (Section 2.3).  are reported in Table 5.

each layer  along a line crossing isopycnals as a composite of segments nearly parallel to isopycnals (Figure 12, upper left panel). The gross lateral density ratio associated to this distribution is in the range of 0.65 – 0.78, with an average of 0.72.  episodes separately, their distribution is encapsulated in single segments  slopes closer to isopycnals (Figure 12, lower panels). Lateral density ratios are in the range of 0.89 – 0.93 for the first episode, 0.82 – 0.98 for the second episode, with an average of 0.91 in both episodes. Given the short timescale of each episode (3-4 months), these ratios are inferred by changes  layer properties attributed to spatial variations, in agreement with the  geographical .  the epicentral region, the distribution extents along lines crossing isopycnals (Figure 12, upper right panel). Lateral density ratios are in the range of 0.74 – 0.83, with an average of 0.80. This ratio is  inter-annual trend in the relative changes  layer temperature and  salinity. As detailed further in Section 4.3, these estimations of lateral density ratios  water mass conversion within thermohaline staircases  driven by two distinct processes, one acting at large spatial scales, the other at large temporal scales.

**3.3 Estimation of nitrate fluxes in presence of thermohaline staircases**

The large-scale distribution  LIW  nitrate profiles collected during  PEACETIME in May-June 2017  increase  along its pathway from the Ionian Sea to the Algerian Basin. The LIW enrichment is particularly significant in the Tyrrhenian Sea, with an increase  2 µmol/kg in nitrate concentration between the eastern Tyrrhenian  and the southwestern Sardinian  (Figure 13) Young LIW are found in the Ionian  salinity  than 38.9) at 200 dbar depth, whereas the

LIW  are found in the Algerian station (salinity 38.6 at 450 dbar). The three  stations have transitional properties,  38.7 in salinity southwest Sardinia. Nitrate concentrations follow an inverse  salinity: profiles at the five stations are clearly differentiated below 250 dbar, showing inflow of low nutrient waters from the Ionian Sea to the eastern Tyrrhenian (same nitrate values at 450 dbar) and their progressive enrichment until the Algerian Basin.  the central Tyrrhenian Sea and  Algerian Basin where thermohaline staircases  (Figure 14, left panel), the sunlit surface layers were equally depleted at this  of the year, while the nutrient stocks were similar in the DW with concentrations of about 8.7 µmol/kg  nitrate. Differences  in the transition zone:  In the  case, nitrate concentrations slightly decrease by 0.7 µmol/kg over a depth range of 500-1500 dbar towards their DW concentrations; while in the  nitrate concentrations increase by 2.3 µmol/kg over a depth range of 500-2000

dbar. A second major difference between the two basins appears on the depth extension of the nitracline. In the Algerian Basin, the base of the nitracline is  LIW (350-500 dbar),  nitrate concentrations reach their maximum value

 In the Tyrrhenian Sea instead,  to 250 dbar: nitrate concentrations reach a local maximum above LIW,  slightly decrease inside the LIW core (salinity maximum at 400 dbar).

 the "S-shape" delineated by the nitrate profile between LIW and the nitracline, mark the isopycnal inflow inside the Tyrrhenian Sea of the nutrient-poor waters from the Ionian Sea.  float 6901769 documented this feature during the two years preceding the cruise.  (Figure 14, right panel). The S-shape appears as a persistent feature in the southwest sector of the basin, with variable  depending on the location. It is pronounced in the southeasternmost profiles of the distribution , with nitrate concentrations  about 4 µmol/kg along isopycnal 29. Moving west towards the Sardinian Channel, the  5.6 µmol/kg  in the central  minimum increases to 6 µmol/kg . Along the eastern Sardinian coast, the  a slight inflexion at 6.5 µmol/kg .  the Ionian water inflow gradually impacts the nutrient  above LIW. This will be further discussed  to the LIW circulation pathway in Section 4.4.

In  to quantify the roles played by thermohaline staircases or Ionian-Algerian inflows on  LIW , the vertical transfers of nitrates  estimated  three distinct : (i) the central Tyrrhenian station of the cruise PEACETIME, with four casts at the same location providing a  density and nitrate concentration,  measured with high accuracy and  vertical resolution; (ii) the Algerian  in which spatial and temporal inhomogeneities are  affecting only the surface layer; (iii)  float, with a  range  (see Figure 5),  less accurate nitrate measurements (Section 2.2).  the two first cases, diapycnal nitrate fluxes are computed using parameterizations detailed in Section 2.4; stages of computation and results are reported in Table 6.  only diapycnal fluxes above LIW  estimated using the  nitrate flux has been  over 98% of profiles, yielding 241 +/- 93 µ
[revised manuscript text omitted]